# Camellia 🌸: Benchmarking Cultural Biases in LLMs for Asian Languages

## Abstract

As Large Language Models (LLMs) gain stronger multilingual capabilities, their ability to handle culturally diverse entities becomes crucial. Prior work has shown that LLMs often favor Western-associated entities in Arabic, raising concerns about cultural fairness. Due to the lack of multilingual benchmarks, it remains unclear if such biases also manifest in different non-Western languages. In this paper, we introduce Camellia, a benchmark for measuring entity-centric cultural biases in nine Asian languages spanning six distinct Asian cultures. Camellia includes 19,530 entities manually annotated for association with the specific Asian or Western culture, as well as 2,173 naturally occurring masked contexts for entities derived from social media posts. Using Camellia, we evaluate cultural biases in four recent multilingual LLM families across various tasks such as cultural context adaptation, sentiment association, and entity extractive QA. Our analyses show a struggle by LLMs at cultural adaptation in all Asian languages, with performance differing across models developed in regions with varying access to culturally-relevant data. We further observe that different LLM families hold their distinct biases, differing in how they associate cultures with particular sentiments. Lastly, we find that LLMs struggle with context understanding in Asian languages, creating performance gaps between cultures in entity extraction.

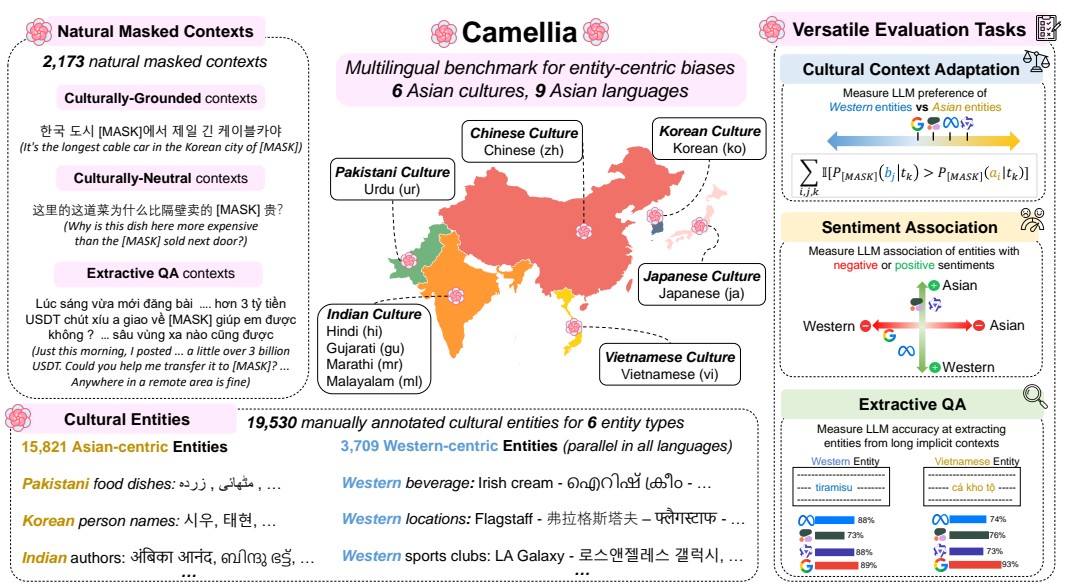

Figure 1: We construct Camellia, a benchmark to measure cultural biases for six Asian cultures, covering nine languages. Camellia provides 2,173 naturally-occurring masked contexts categorized into: culturally-grounded, culturally-neutral, and extractive QA. Camellia also provides 19,530 culturally relevant entities that contrast the respective Asian cultures vs. Western culture across six different entity types that exhibit cultural variation. The masked contexts and entities in Camellia enable the measurement of cultural biases in LLMs via versatile task setups.

# 1 INTRODUCTION

Large Language Models (LLMs) have rapidly integrated into modern technology, serving users from diverse cultures (Adilazuarda et al., 2024). Among the vast range of text they process, LLMs frequently encounter entities such as people's names, locations, or food dishes, which are pervasive in text corpora (Wolfe & Caliskan, 2021; Pawar et al., 2025a) and often appear in user prompts (Li et al., 2024a; Wang et al., 2025). Importantly, entities carry cultural associations, making it essential for LLMs to handle culturally diverse entities fairly. However, past work has shown that these cultural associations can significantly influence LLMs, leading to biased behaviors (An et al., 2024; Wan et al., 2023). The recent study of Naous et al. (2024) demonstrated how such biases manifest when testing LLMs in Arabic, where models showed better performance on entities associated with Western culture compared to those linked to Arab culture. A natural question is *whether similar LLM cultural biases would also manifest in other non-Western languages.*

To this end, we introduce Camellia (**C**ultural **A**ppropriateness **Me**asure Set for **LLMs** **i**n **A**sian Languages), a benchmark for measuring entity-centric cultural biases in 9 non-Western languages spoken in the Asian continent: Chinese (zh), Japanese (ja), Korean (ko), Vietnamese (vi), Urdu (ur), Hindi (hi), Malayalam (ml), Marathi (mr), and Gujarati (gu), covering 6 distinct cultures in Asia (see Figure 1). Following the data curation process outlined in CAMeL (Naous et al., 2024), we undertook a year-long collaboration with native speakers to collect and annotate 19,530 cultural entities across six entity types contrasting Asian and Western cultures (§3.1). We also curate 2,173 naturally occurring masked contexts for entities spanning all nine languages (§3.3). Moreover, we provide English translations for each entity and masked context in Camellia, enabling direct cross-lingual comparisons for testing LLMs in English vs the respective Asian language.

In summary, we make the following key contributions:

- We introduce Camellia, a benchmark to study entity-centric cultural biases in LLMs for 9 Asian languages, covering 19,530 entities and 2,173 masked contexts annotated by native speakers, enabling us to benchmark recent multilingual LLMs across three task setups: cultural adaptation, sentiment association, and extractive QA.

- We show how **LLMs can struggle to adapt to the cultural contexts of the Asian cultures in Camellia**, assigning higher likelihood for Western entities in 30-40% of cases, even when inappropriate to the context (§4.1).

- We reveal that **different model families can also display their own distinct biases** in sentiment association, where Qwen shows a higher tendency of associating Asian entities with positive sentiment compared to Western entities, whereas the Llama and Gemma models show the opposite trend (§4.2).

- We show how **LLMs still lack the ability to efficiently grasp context in the Asian languages we tested, impacting their cultural fairness in entity extraction**. When tasked with extracting entities from paragraphs, we observed large accuracy gaps in LLMs when entities in the same text were associated with different cultures. In contrast, these gaps were minimal when testing LLMs on the English translations of contexts and entities, where performance is stable regardless of an entity's cultural association (§4.3).

# 2 RELATED WORK

**Multilingual Cultural Evaluation of LLMs.** The rapid deployment of LLMs has sparked recent interest from the research community in their cultural evaluation (Liu et al., 2025; Qadri et al., 2025a;b; Singh et al., 2025), resulting in the release of various benchmarks (Pawar et al., 2025b). Past work has introduced several question-answering datasets that evaluate models on open-ended culture-specific questions (Chiu et al., 2024b;a; Myung et al., 2024). Other works have focused on constructing knowledge bases to evaluate specific cultural domains such as culinary practices (Palta & Rudinger, 2023; Zhou et al., 2024) or social norms (Rao et al., 2024; Fung et al., 2024). Multilingual resources have also been introduced to evaluate LLMs on geo-diverse facts (Yin et al., 2022; Keleg & Magdy, 2023; Dammu et al., 2024), regional exam questions (Romanou et al., 2024; Singh et al., 2025), and questions on local norms sourced from native speakers (Guo et al., 2025; Alwajih et al., 2025). A few studies have also introduced benchmarks for multilingual multi-modal

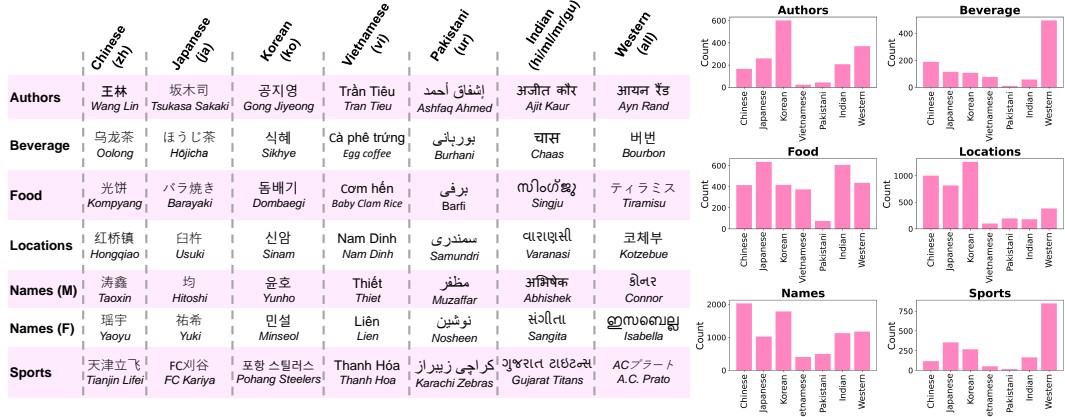

| | Chinese (zh) | Japanese (ja) | Korean (ko) | Vietnamese (vi) | Pakistani (ur) | Indian (hi/ml/mr/gu) | Western (all) |
|---|---|---|---|---|---|---|---|
| **Authors** | 王林 *Wang Lin* | 坂木司 *Tsukasa Sakaki* | 공지영 *Gong Jiyeong* | Trần Tiêu *Tran Tieu* | إشفاق أحمد *Ashfaq Ahmed* | अजीत कौर *Ajit Kaur* | आयन रैंड *Ayn Rand* |
| **Beverage** | 乌龙茶 *Oolong* | ほうじ茶 *Hōjicha* | 식혜 *Sikhye* | Cà phê trứng *Egg coffee* | بورہانی *Burhani* | चास *Chaas* | 버번 *Bourbon* |
| **Food** | 光饼 *Kompyang* | バラ焼き *Barayaki* | 돔배기 *Dombaegi* | Cơm hến *Baby Clam Rice* | برفی *Barfi* | സിംഗ്ജു *Singju* | ティラミス *Tiramisu* |
| **Locations** | 红桥镇 *Hongqiao* | 臼杵 *Usuki* | 신암 *Sinam* | Nam Dinh *Nam Dinh* | سمندری *Samundri* | वाराणसी *Varanasi* | 코체부 *Kotzebue* |
| **Names (M)** | 涛鑫 *Taoxin* | 均 *Hitoshi* | 윤호 *Yunho* | Thiết *Thiet* | مظفر *Muzaffar* | अभिषेक *Abhishek* | 커너 *Connor* |
| **Names (F)** | 瑶宇 *Yaoyu* | 祐希 *Yuki* | 민설 *Minseol* | Liên *Lien* | نوشین *Nosheen* | संगीता *Sangita* | ഇസബെല്ല *Isabella* |
| **Sports** | 天津立飞 *Tianjin Lifei* | FC刈谷 *FC Kariya* | 포항 스틸러스 *Pohang Steelers* | Thanh Hóa *Thanh Hoa* | کراچی زیبراز *Karachi Zebras* | ગુજરાત ટાઇટન્સ *Gujarat Titans* | ACプラート *A.C. Prato* |

Figure 2: Example per entity type and statistics of respective Asian entities per culture and Western entities in Camellia. Western entities are parallel for all 9 languages while Indian entities are parallel in all Indian languages (§3.1). Camellia also provides an English translation for each entity.

cultural evaluations, such as the recognition of culture-specific traditions (Romero et al., 2024) or food dishes (Winata et al., 2024; Lavrouk et al., 2025; Li et al., 2024b). Less work has evaluated the sensitivity of LLMs to entities that exhibit cultural variation (Nikandrou et al., 2025; Zhao et al., 2025; Naous et al., 2024; An et al., 2024; Nghiem et al., 2024; Arora et al., 2025). Our work introduces Camellia, a benchmark to measure entity-centric cultural biases in 6 non-Western cultures in Asia and 9 diverse Asian languages. Camellia includes 2,173 natural masked contexts constructed from social media posts and 19,530 cultural entities extracted from Wikidata and mC4 web-crawls with manual annotation.

**LLM Biases in Asian Languages.** There exist various studies that introduce multilingual resources for measuring biases in LLMs, which cover languages spoken in the Asian continent. Much of the prior work probe LLMs for demographic biases using manually written templates (e.g.; *Everyone hates {attribute}*) (Levy et al., 2023), focusing on attributes such as gender (Ding et al., 2025; Vashishtha et al., 2023; Kaneko et al., 2022), race (Costa-jussà et al., 2023), religion (Rinki et al., 2025), age (Zhao et al., 2023), and more (Lan et al., 2025; Hsieh et al., 2024). Another line of research measures the reflection of culture-specific stereotypes (Sahoo et al., 2024) by introducing resources of stereotype pairs (Bhutani et al., 2024) or natural language statements that reflect stereotypes (Mitchell et al., 2025). Other works have adapted existing English benchmarks (Parrish et al., 2021) for measuring stereotypes in QA model outputs into Chinese (Huang & Xiong, 2023), Japanese (Yanaka et al., 2025), and Korean (Jin et al., 2024). Monolingual resources have been introduced to measure moral bias in Chinese (Hämmerl et al., 2022), and political bias in Urdu (Nadeem et al., 2025). Different from existing research, our work focuses on measuring biases in LLMs when handling Asian vs Western-centric entities, covering 6 Asian cultures and 9 Asian languages.

## 3 CONSTRUCTING CAMELLIA

This section describes the process of constructing the Camellia benchmark. First, we outline our methodology for collecting culturally-relevant entities across nine different Asian languages (§3.1). We then discuss some language-specific challenges faced when collecting data for a diverse set of cultures and languages which required special design decisions (§3.2). We then describe how we collect naturally-occurring masked contexts for entities, which enable testing for entity-centric cultural biases in LLMs across versatile setups (§3.3).

### 3.1 COLLECTING CULTURAL ENTITIES

Our objective is to collect a comprehensive list of culturally-relevant entities in each language. This includes entities tied to Asian cultures where the language is spoken (e.g., entities associated with

Pakistani culture in Urdu, Chinese culture for Chinese, etc.) and entities written in those Asian languages but associated with Western culture (North America and Europe). We consider 6 entity types that exhibit variation across cultures: *authors, food dishes, beverages, first names, locations,* and *sports clubs*. To collect entities, we follow the procedure described in the CAMeL benchmark (Naous et al., 2024), which leverages the multilingual Wikidata knowledge base and performs pattern-based extraction on web-crawled data. Figure 2 shows the statistics of Asian-centric and Western entities that we collect and annotate for each language in `Camellia`.

**Defining Asian vs. Western cultures.** For natives of the Asian cultures that we study, there exists a clear distinction between entities that are associated with their native Asian culture and entities that are typically viewed as Western by those cultures. For example, native Chinese associate the first name "*Weili*" with Chinese culture and the first name "*Valentina*" with Western culture. Similarly, native Pakistanis associate the dish "*Nihari*" with Pakistani culture and the dish "*Lasagna*" with Western culture. We follow this natural phenomena of entity cultural association to distinguish between entities that are native to each Asian culture in Camellia from Western-associated entities.

Western culture encompasses a vast range of countries across different continents, for which the entities of these countries appear sparsely in the different Asian languages we study. From the perspective of these Asian cultures, Western culture is generally understood to include North America and Europe. We therefore limit our Western entities to countries in North America (i.e., United States, Canada, Mexico) and Europe. We group all entities from these countries under a broad Western culture, rather than analyzing each country separately, which simplifies the design of our benchmark. We note that this excludes some Western-dominated regions such as Australia, which is a limitation that we discuss at the end of our paper. We also report the country-wise distribution of the Western entities in `Camellia` in Appendix A.

**Extracting Entities from Wikidata.** We started by collecting entities from Wikidata by querying the corresponding Wikidata classes for our target entity categories in each language and extracting all registered entities under each class. We found the coverage in Wikidata to be generally sufficient for *authors*, *locations*, and *sports clubs* for all languages. However, the coverage for the other entity types (*food dishes, beverages, names)* was much less extensive and varied by language. As of 2024, we observed that higher-resource languages had a sizable amount of entities in Wikidata (e.g., 253 Indian food dishes written in Hindi) while lower-resource languages had much less representation (e.g., only 24 Indian food dishes in Malayalam, 37 Pakistani names in Urdu, etc.).

**Pattern-based Extraction from Web-Crawls.** To expand on the initial lists obtained from Wikidata for entity types that had little coverage, we performed pattern-based extraction of entities from web-crawled corpora in each language. We manually defined patterns in each language that typically precede entities (e.g., *brother/sister named* ⎯⎯ for first names, *recipe of* ⎯⎯ for food dishes, etc.). Using the patterns, we scanned through each language's partition in the mC4 web-crawl corpus (Xue et al., 2021) and extracted unigrams and bigrams that appeared after a detected pattern. We also accounted for gender inflections if required. This resulted in 5k-10k extractions in each type and language, which were then manually filtered to remove irrelevant extractions and select culturally-relevant entities. Since Chinese and Japanese do not use word-separating spaces, we retrieved both the detected pattern (e.g., "喝", which means "*to drink*") and up to ten surrounding characters in these languages, and then prompted `GPT-4o-mini` to extract the entity from the captured characters, if any were mentioned. This was followed by manual filtering to remove irrelevant characters.

**Annotation by Native Speakers.** The annotation was conducted by nine different authors in total, each a native speaker of one of the 9 Asian languages in `Camellia`. This involved manual filtering of the Wikidata and mC4 extractions to identify culturally relevant entities and remove irrelevant ones. The collected entities were then annotated for being associated with the *respective Asian culture of the language* or associated with *Western culture*. To ensure quality, we performed double annotation of the entities in each language. The second annotators consisted of undergraduate or master's students hired for zh, ja, ko, hi, ml, mr, and gu; and native speaker volunteers for vi and ur. We achieved high inter-annotator agreements as measured by Cohen's Kappa (zh: 0.85, ja: 0.78, ko: 0.92, vi: 0.80, ur: 0.88, hi: 0.94, ml: 0.83, mr: 0.93, gu: 0.97). The disagreements were then resolved in an adjudication step to decide the final label. We report the detailed annotation guidelines for each entity type in Appendix A.

**Translating Entities to English.**   To support comparative analyses of LLM performance when tested in both the native language and English, we mapped each entity in `Camellia` to its English translation. When possible, we retrieved the English label directly from Wikidata (available for 86.58% of Wikidata-sourced entities). For entities without an English label and ones extracted from mC4, we manually searched for their most commonly used English transliterated form found online, ensuring that the translations reflect how entities appear in real-world usage.

**Parallelizing Western Entities.**   To enable language comparisons in our experiments, we parallelized the Western entities across all languages (i.e., each entity has a written version in every language). For *authors*, *locations*, and *sports clubs*, we constructed their parallel Western sets directly from Wikidata by extracting the entities of each Western country (North America and Europe) that had a written form in at least 6 of the languages. A lot of these Western entities did not have written versions in Wikidata in low-resource languages (`ur`, `ml`, `gu`, and `mr`). For those languages, we manually filled in their missing translations.

For the other types of *food*, *beverage*, and *names*, Western entities were collected independently in each language via pattern-based extractions mC4. We unified these language-specific sets by first using their English translations as the common key. Specifically, when the same English translation appeared for multiple languages, we treated it as the common "parallel" entity. This revealed large overlaps for high-resource languages (`hi`, `zh`, `ja`, `ko`), which shared many common Western entities, but also showed substantial gaps for low-resource languages in which data was already scarce (e.g., 1k–1.5k food entities needed to be translated to *ur*). To minimize translation effort while ensuring quality, we randomly sampled 500 unified entities per type and, with the help of annotators, manually completed the missing entries by translating them from English into their languages.

**Parallelizing Entities in Indian Languages.**   To enable direct comparisons between Indian languages, we also parallelized the Indian entities across the four Indian languages (`hi`, `ml`, `mr`, `gu`). Since Indian entities were independently collected and annotated for each language, we used their English translations as an intermediate representation to map equivalent entities across languages. Annotators then manually translated the missing gaps from English. The majority of Indian cultural entities were initially collected in `hi`, being the most resource-rich Indian language. In contrast, manual translation efforts were mostly required to map entities into `ml`, `mr`, and `gu`.

## 3.2 LANGUAGE-SPECIFIC CHALLENGES

We now discuss some of the entity-specific challenges we encountered while constructing `Camellia`. These challenges stem from diverse linguistic and cultural factors that shaped several of our dataset design choices. Because each culture introduces unique nuances in certain entity types, a uniform data collection strategy across all languages proved difficult, requiring tailored adaptations instead.

**Entity naming conventions can be subject to temporal change.**   In Korea, China, and Japan, modern names differ significantly from older ones (Barešová & Janda, 2023). For instance, many Korean feminine names in the mid-20th century included elements like '*suk*' (숙) or '*mi*' (미), which symbolize purity and beauty, respectively. In contrast, contemporary names like '*Seo-yun*' (서윤) or '*Ji-woo*' (지우) reflect trend-driven preferences. Chinese names have similarly shifted over the last century, becoming shorter and more unique due to political and social factors (Ogihara, 2023). Such temporal changes can make it challenging to collect entities that are representative today. For example, the Korean, Chinese, and Japanese first names listed on Wikidata are mostly outdated names with little to no contemporary usage. To more accurately reflect modern naming conventions, we used recent governmental statistical reports in Korea[1] and China[2]. For Japanese, due to a lack of similar reports, we used a popular name generator[3] to generate Japanese first names. All names were then verified to be valid by our native annotators.

**Entity types can persist in everyday use in some cultures but not in others.**   The CAMeL benchmark (Naous et al., 2024) initially included a clothing entity type contrasting traditional Arab

---

[1] https://efamily.scourt.go.kr
[2] 2021 National Name Report
[3] https://namegen.jp

clothing with Western attire. However, extending this to other non-Western cultures proves challenging. For instance, in Pakistani culture, traditional garments such as the "*shalwar kameez*" remain a common part of everyday attire (Ranavaade & Karolia, 2017). In contrast, in many other Asian societies, including China and Japan, traditional clothing like the "*hanfu*" is now generally reserved for special occasions. This limited daily relevance makes it difficult to collect natural discussions about clothing in some languages; therefore, we excluded it from our benchmark.

**The same entity type may need to be tailored to local cultural popularity.** The same entity type can carry different meanings depending on the culture, reflecting what people care about and commonly discuss. This is illustrated by the sports clubs category in Camellia. We focused on sports that have a strong imprint in each culture. In Pakistan and India, for example, cricket holds significant importance and even influences political discourse between the two countries (Chakraborty, 2022); accordingly, we collected cricket clubs as the sports club entities for these cultures. In contrast, across much of East and Southeast Asia, we focused on football as one of the most widely followed sports (Connell, 2018). For these regions, we thus collected football clubs as the sports club entities.

### 3.3 COLLECTING NATURAL MASKED CONTEXTS

To evaluate whether LLMs can distinguish between entities associated with each Asian culture vs. those associated with Western cultures, Camellia provides 2,173 naturally-occurring masked contexts for entities derived from natural discussions by native speakers on X (formerly Twitter). We source our contexts from X for all languages. For Chinese, however, we use the Weibo and Xiaohongshu platforms instead, since X is officially blocked in China.

Following CAMeL (Naous et al., 2024), we collected short contexts that are uniquely suited for the entities associated with each Asian culture, enabling us to assess LLM cultural adaptation. We also collected neutral contexts where entities from any culture were appropriate, helping determine the default inclinations of models in the absence of clear cultural cues. Additionally, we constructed longer contexts that reference entities more implicitly, presenting a challenging setup for testing models at entity identification in an extractive QA format. Accordingly, the masked contexts in Camellia are split into three types: **(1)** culturally-grounded (Camellia-Grounded), **(2)** culturally-neutral (Camellia-Neutral), and **(3)** extractive QA contexts (Camellia-QA).

**Contexts for Evaluating Cultural Adaptation.** To construct Camellia-Grounded, we searched using two types of search queries: randomly sampled Asian entities (e.g., [Indian entity], [Japanese entity]), and manually designed patterns that mention a culturally-relevant entity (e.g., the [Chinese] city of, the [Indian] dish, etc.). We then manually inspected the retrieved tweets to identify ones that provide suitable cultural contexts (i.e. contexts where only an entity associated with the respective Asian culture can be placed). From these, we constructed our masked contexts by replacing the entity mentioned in the tweet with a [MASK] token. Similarly, to construct neutral contexts (Camellia-Neutral), we identified tweets where entities from any culture would be appropriate as [MASK]. Further, we annotated each context with one of three sentiment labels: *positive*, *negative*, or *neutral*. This helps evaluate whether substituting the [MASK] token with the respective Asian or Western entities changes the sentiment predicted by LLMs (§4.2).

**Contexts for Extractive QA.** In addition to the contexts used to evaluate cultural adaptation in LLMs, we constructed longer, paragraph-level contexts in which entities are mentioned implicitly. These longer contexts enable a challenging evaluation setup for entity extraction, as they require understanding the underlying context to identify the entity. We follow the same keyword search strategy to identify such contexts, and replace the mentioned entity with the [MASK] token. Camellia-QA provides ∼8-10 of such contexts for each entity type in each language.

**Parallelizing Indian Contexts.** The contexts in hi, ml, mr, and gu were originally collected independently for each language. To enable comparisons across these Indian languages, we parallelized them by first translating the contexts into English and then into the other Indian languages.

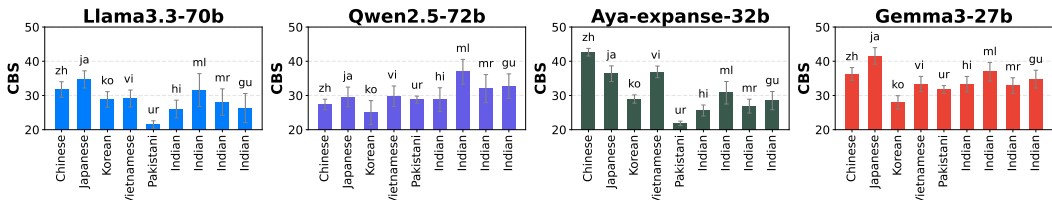

Figure 3: Average **C**ultural **B**ias **S**core (CBS) (↓) across entity types achieved by LLMs on culturally-grounded contexts (`Camellia-Grounded`) for each Asian language. LLMs can struggle to generate the appropriate Asian entities in each culture, assigning better likelihood to Western entities 30-40% of the time. See results per entity type in Appendix C.1.

## 4   ARE CULTURAL BIASES CONSISTENT ACROSS LANGUAGES AND LLMS?

We leverage the cultural entities and masked contexts in `Camellia` to investigate whether cultural biases in LLMs are persistent across languages and LLMs. We experiment with four recent LLMs with multilingual capabilities: **Llama3.3-70b** (Grattafiori et al., 2024), **Qwen2.5-72b** (Yang et al., 2025), **Aya-expanse-32b** (Dang et al., 2024), and **Gemma3-27b** (Team et al., 2025). We test LLMs in three setups: cultural adaptation (§4.1), sentiment association (§4.2), and extractive QA (§4.3).

### 4.1   CULTURAL CONTEXT ADAPTATION

We first analyze the ability of LLMs to adapt to different Asian cultural contexts by analyzing their assigned likelihood for the respective Asian vs Western entities as [MASK] token fillings.

**Cultural Bias Score (CBS).** We use the CBS designed by Naous et al. (2024) to measure the level of Western bias in an $\text{LLM}_\theta$. CBS is a likelihood-based measure that computes the percentage of an LLM's preference for Western entities over Asian ones within the same cultural context. Given an entity type $D$, two type-specific sets of respective Asian entities $A = \{a_i\}_{i=1}^N$ and Western entities $B = \{b_j\}_{j=1}^M$, and a masked context $c_k$, we compute $\text{CBS}_D(\text{LLM}_\theta, A, B, c_k)$ per language as:

$$\text{CBS}_D(\text{LLM}_\theta, A, B, c_k) = \frac{1}{N \times M} \sum_{i=1}^N \sum_{j=1}^M \mathbb{1}[P_{\texttt{[MASK]}}(b_j|c_k) > P_{\texttt{[MASK]}}(a_i|c_k)], \quad (1)$$

where $P_{\texttt{[MASK]}}$ is the LLM's probability of an entity filling the [MASK] token. For entities tokenized into multiple tokens, we take the product of the conditional probabilities of each token. For a set of prompts $C = \{c_k\}_{k=1}^K$, the CBS per entity type for an LLM is computed by averaging over all $c_k \in C$. An LLM is considered more Western-biased as its CBS gets close to 100%.

**Results.** Figure 3 shows the average CBS across entity types achieved on the culturally-grounded contexts of each culture when tested in each language. We observe the following key insights:

**LLMs can struggle to distinguish Asian vs. Western entities.** Since the contexts we test on are grounded in each Asian culture (only entities associated with the specific Asian culture are appropriate for filling the [MASK]), models should always assign higher likelihood to the native Asian entities in those contexts, and the CBS is expected to be low (closer to the 0-5% range (Naous et al., 2024)). However, in most cases, we observe the CBS to be in the 30-40% range. This highlights many situations where LLMs struggle to differentiate between Asian and Western entities, assigning a better likelihood to Western entities despite being inappropriate to the context.

**Are models sensitive to cultural grounding?** We further analyze if performance changes when testing on the contexts that are culturally neutral (i.e., any entity is an appropriate [MASK] filling in the context). The results are summarized in Figure 4, which shows that CBS scores are higher when contexts are neutral, with LLMs becoming more likely to generate Western entities. However, in the majority of cases, the scores still remain very close to when contexts are culturally grounded.

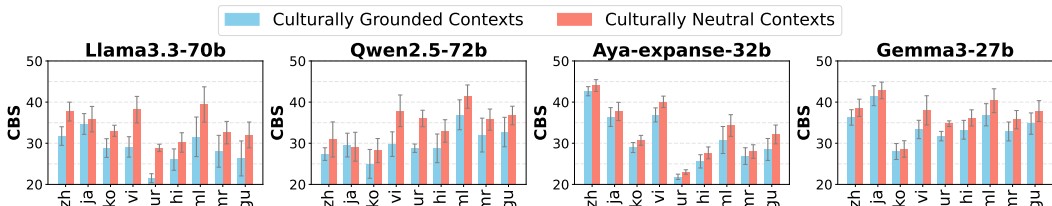

Figure 4: Average CBS across entity types on culturally-grounded contexts vs culturally-neutral contexts. LLMs show more preference towards Western entities in culturally-neutral contexts (higher CBS). CBS scores are lower in culturally-grounded contexts, yet remain close to the neutral case.

This suggests a lack of sensitivity to cultural contexts in LLMs, whereby their ability to select the appropriate entities at generation time is not greatly impacted by cultural grounding.

**Adaptation performance can vary by LLM family.** Noticeable differences can be seen in the performance of LLM families developed in different regions. Specifically, we find that the Qwen2.5-72b model that is developed by China-based Alibaba performs the best on Chinese, Japanese, and Korean, compared to the rest of the models. One likely reason for such a gap could be more access to culturally relevant pre-training data in those languages, enabling the model to learn cultural associations that others would miss. This highlights the importance of data provenance in shaping the cultural competence of LLMs. Moreover, this corroborates the results of past work that shows a better ability of Qwen models at answering questions specific to Chinese culture (Guo et al., 2025). We also find that having more representation of the script of a language in the model tokenizer leads to improved performance (see tokenizer analysis in Appendix C.1).

**Adaptation ability for the same culture can vary by resource availability.** In the Indian setting, performance varied based on the resource availability of languages. Models performed relatively better when tested in Hindi but struggled more when tested in lower-resource languages as Malayalam, Marathi, and Gujarati. Notably, this trend is consistent across all models, reflecting similar access to training data proportions for those languages. In practice, this makes the adaptation ability of LLMs to Indian contexts skewed towards Hindi, privileging one linguistic community over others.

## 4.2 SENTIMENT ASSOCIATION

We examine whether LLMs subtly associate entities from Asian or Western cultures with specific sentiments by analyzing their behavior on sentiment analysis.

**Setup.** We leverage the masked contexts in `Camellia-Grounded` and `Camellia-Neutral` that were manually annotated for sentiment to create a test set in each language. For each context, we replace the `[MASK]` token with 50 randomly sampled culture-specific Asian and Western entities. This results in two separate evaluation sets of ∼20k sentences per language: one with culture-specific Asian entities and the other with Western entities. Importantly, the contexts remain the same across both sets, allowing us to isolate the effect of entity cultural association on changes in the LLMs' predictions. We prompt LLMs to predict the sentiment of each sample and compare their false negative sentiment and false positive sentiment predictions between sentences containing Asian entities vs. Western entities. Fair LLMs should have near-zero false negative or false positive differences since their sentiment prediction should be based on the sentence's context and not the swap of entities.

**Results.** Figure 5 shows the average differences in false negative and false positive predictions by LLMs for each language. We observe that **sentiment associations vary significantly across different LLMs**. For instance, Llama and Gemma exhibit a stronger tendency to associate Western entities with negative sentiment, whereas Qwen and Aya often associate Asian entities with positive sentiment, particularly in Indian languages. These results highlight how current LLMs can be sensitive to cultural associations of entities when used as classifiers - a critical consideration for different use cases of LLMs, such as content moderation, where these biases can lead to unfair decisions (Garg et al., 2023). LLM-specific sentiment biases are likely a reflection of differences in their

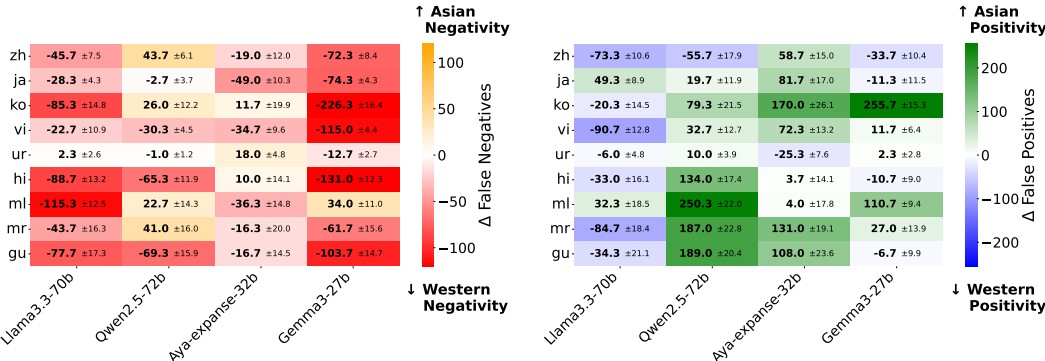

Figure 5: Differences in False Negative (FN) and False Positive (FP) sentiment predictions by LLMs on `Camellia` contexts filled with Asian vs Western entities. Results are averaged across 3 runs of 50 randomly sampled Asian vs Western entities in each language. Llama and Gemma tend to associate Western entities with negativity, while Qwen and Aya tend to associate Asian entities with positivity.

training data, where models can learn spurious associations when cultural entities appear frequently in positive or negative contexts.

## 4.3 EXTRACTIVE QA

We now analyze the ability of LLMs to extract entities from paragraph-long contexts. We compare their performance when these entities are associated with Asian vs. Western cultures.

**Setup.** Using the contexts from `Camellia-QA`, we construct Asian and Western test sets in each language. For each context, we replace the [MASK] with 50 randomly sampled entities, in a similar manner to our earlier experiment for sentiment association (§4.2). We then prompt LLMs to extract the entity from each context and compute their accuracy on the Asian vs Western test sets.

**Results.** Figure 6 shows the average accuracy achieved by LLMs for each Asian language. We observe a consistent trend where **LLMs generally achieve higher accuracy in extracting entities associated with each Asian culture rather than Western-associated entities**. There are a few cases showing the opposite behavior, specifically in Vietnamese and Urdu, where the Llama and Qwen models achieve better accuracy on Western entities than Pakistani and Vietnamese entities.

To compare whether these gaps are also observed in English, we test all

| Culture | Llama3.3-70b | | Qwen2.5-70b | | Aya-expanse-32b | | Gemma3-27b | |
|---|---|---|---|---|---|---|---|---|
| | Asian | English | Asian | English | Asian | English | Asian | English |
| Chinese | -1.32 | 0.30 | 0.43 | -2.84 | 2.84 | -5.83 | -1.36 | -5.63 |
| Japanese | 7.55 | 2.72 | 18.87 | 4.53 | 8.84 | -0.73 | 16.40 | -3.22 |
| Korean | 9.69 | 0.66 | 16.47 | -2.49 | 13.94 | 1.43 | 7.94 | 2.54 |
| Vietnamese | -13.53 | 1.95 | -14.33 | -3.61 | 2.83 | -1.88 | 4.15 | 1.65 |
| Pakistani | -4.71 | 10.54 | -4.99 | 12.16 | 0.12 | 4.54 | 21.11 | 4.54 |
| Indian (hi) | 10.05 | 6.71 | 3.63 | 10.67 | 11.54 | 1.07 | 6.81 | 3.25 |
| Indian (ml) | 13.15 | — | 4.22 | — | 10.93 | — | 9.01 | — |
| Indian (mr) | 11.07 | — | 1.68 | — | 12.64 | — | 3.50 | — |
| Indian (gu) | 14.44 | — | 6.02 | — | 12.89 | — | 6.54 | — |

Table 1: ΔAccuracy on extractive QA between Western and Asian entities when testing models on parallel data in the respective Asian language of each culture vs. in English. Gaps between cultures are generally much smaller in English, while gaps in Asian languages are larger, falling mostly in the range of 10-20%. See detailed results in Appendix C.3.

models on the parallel English data for each culture. Table 1 compares the QA accuracy difference between Asian and Western entities when testing models in the respective Asian language of each culture vs. English. We find that gaps between cultures in English are much smaller, ranging mostly between 1% and 5%, with no clear trend of superior performance on one culture. Yet, gaps in Asian languages are much larger, reaching a 12%-20% range in most cases, with the exception of Chinese, where gaps were minimal. These results show that **LLMs still lack a robust ability to grasp implicit contexts in most of these non-English languages we tested on, creating large performance gaps between different cultures**. As noted in past work, these gaps may be due to a lack of representation of certain cultural entities in pre-training, where models may get lost when encountering

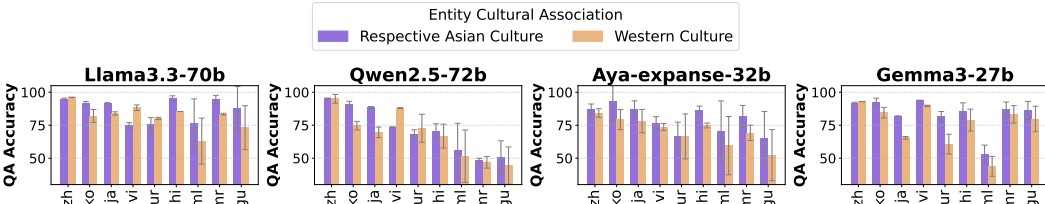

Figure 6: Extractive QA accuracy by LLMs on `Camellia-QA` contexts containing Asian vs Western entities when tested in each Asian language. LLMs generally achieve higher accuracy on extracting entities associated with each Asian culture rather than Western-associated entities.

entities as rarely seen tokens (Li et al., 2024a). This may also be a result of linguistic phenomena where LLMs struggle to distinguish multi-sense words that overlap with cultural entities (Naous & Xu, 2025).

## 5 CONCLUSION

We introduced Camellia, a comprehensive benchmark for evaluating entity-centric cultural biases in 9 Asian languages across 6 distinct cultures. Through systematic analyses, we demonstrated that current multilingual LLMs exhibit various types of cultural biases in these non-Western languages. Models showed struggles in adapting to Asian cultural contexts when tested in their native languages. Our experiments also revealed divergent sentiment associations across model families and performance gaps between cultures in entity extraction. Notably, these issues were greatly reduced when testing on the parallel contexts and entities in English, highlighting the nuanced challenges presented by different languages. We hope that Camellia will serve as a valuable resource and testbed to support future research aimed at developing more culturally aware and fair multilingual LLMs, improving their usability across diverse linguistic and cultural settings.

## LIMITATIONS

In Camellia, we defined the broad Western culture as countries that are exclusively in North America and Europe. However, there are many countries in other geographical regions where Western culture dominates such as Australia, New Zealand, and South American countries, that were excluded from our definition, which is a limitation of our benchmark. Our focus was to explore cultural biases in LLMs when contrasting entities associated each Asian culture we study against those associated with the broad Western culture. We thus followed the view of North America and Europe as representing Western culture, and for which data could be more easily collected in the Asian languages we study. We hope that future work can expand on our set of Western entities to include more representation from these other regions to enable more fine-grained comparisons to Western countries.

## ETHICS STATEMENT

While collecting data from naturally-occurring tweets to construct the masked contexts in Camellia, we discarded any tweets during our search that included offensive or toxic language, hate speech, stereotypes, or included any personally identifiable information. Data collection was done through a manual process by searching on the X, Weibo, and Xiaohongshu platforms, without the use of any automated scraping. We do not share the raw social media posts but modified versions where cultural entities are replaced by a [MASK], which can be used for research purposes. The Camellia benchmark is constructed for the purpose of testing cultural biases in LLMs and enabling future research on the development of LLMs that work efficiently and fairly for all entities regardless of the cultural associations they carry.

## REPRODUCIBILITY STATEMENT

The Camellia benchmark will be made publicly available to the community, which includes the collected entities with their annotations for cultural association and the naturally-occurring masked contexts for all languages. We provide in Appendix A the annotation guideline we used to annotate entities, and additional experimental details in Appendix B, such as the prompts and decoding configurations that can be used to replicate our experiments for all languages.

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

# A  CAMELLIA: ADDITIONAL DETAILS

**Statistics for entities and masked contexts.**  Table 2 shows the number of entities for each language and entity type that we collect and annotate in `Camellia`. Table 3 shows the number of masked contexts that we constructed in each language. We note that fewer contexts could be collected in Urdu due to the low-resource nature of the language, with relatively much less digital presence on social media compared to the rest of the languages.

**Wikidata Classes.**  Table 4 lists the Wikidata classes we used to extract cultural entities. For each language, we identify the relevant country (e.g., India for `hi, ml, gu`, Pakistan for `ur`, Vietnam for `vi`, etc.) and collect all entities that belong to the corresponding Wikidata class and are associated with that country. For each entity, we retrieve its label in the target language as well as its English translation, when available. To collect Western entities, we similarly extract entities for all countries in North America and Western Europe.

| Entity Type | #Cultural Entities | | | | | | |
|---|---|---|---|---|---|---|---|
| | zh | ja | ko | vi | ur | hi/ml/mr/gu | western |
| Authors | 165 | 260 | 602 | 24 | 44 | 207 | 370 |
| Beverage | 189 | 115 | 107 | 77 | 11 | 34 | 497 |
| Food | 415 | 635 | 416 | 374 | 75 | 605 | 436 |
| Locations | 1,000 | 817 | 1,260 | 90 | 196 | 181 | 382 |
| Names (M) | 906 | 503 | 899 | 251 | 334 | 651 | 588 |
| Names (F) | 1,123 | 523 | 886 | 151 | 163 | 563 | 587 |
| Sports | 116 | 354 | 266 | 51 | 17 | 165 | 849 |
| **Total** | 3,914 | 3,207 | 4,436 | 1,018 | 840 | 2,406 | 3,709 |

Table 2: Number of entities for each language and entity type in `Camellia`. Western entities are parallel across all languages. Each entity is also available as an English translation.

| Language | #Masked Natural Contexts | | |
|---|---|---|---|
| | Camellia-Grounded | Camellia-Neutral | Camellia-QA |
| zh | 131 | 126 | 64 |
| ja | 137 | 140 | 60 |
| ko | 150 | 208 | 70 |
| vi | 165 | 192 | 78 |
| ur | 70 | 70 | 58 |
| hi/ml/mr/gu | 215 | 192 | 47 |
| **Total** | 868 | 928 | 377 |

Table 3: Number of masked contexts collected for each language in `Camellia`. Indian contexts are parallel across all Indian languages. Each masked context is also available as an English translation.

| Entity Type | Wikidata Class | Class QID |
|---|---|---|
| Authors | writer | Q36180 |
| | novelist | Q6625963 |
| Beverage | drink | Q40050 |
| Food | food | Q2095 |
| | dish | Q746549 |
| Location | city | Q515 |
| Names (F) | female given name | Q11879590 |
| Names (M) | male given name | Q12308941 |
| Sports Clubs | association football club | Q476028 |
| | cricket team | Q17376093 |

Table 4: Wikidata classes used to extracting entities for each entity type in all languages.

**Lengths of Contexts.** Table 5 shows the average lengths of the contexts in Camellia for each language. We report word length for all languages except for Chinese and Japanese, for which report character length since they do not use spacing.

| Language | Cultural Adaptation Contexts | Extractive QA Contexts |
|---|---|---|
| zh | $37.95_{\pm 8.99}$ | $81.59_{\pm 18.20}$ |
| ja | $58.47_{\pm 16.39}$ | $115.08_{\pm 26.27}$ |
| ko | $9.13_{\pm 4.62}$ | $32.30_{\pm 6.57}$ |
| vi | $38.44_{\pm 13.44}$ | $64.17_{\pm 6.92}$ |
| ur | $15.63_{\pm 6.27}$ | $47.02_{\pm 13.42}$ |
| hi | $21.21_{\pm 12.02}$ | $45.40_{\pm 13.05}$ |
| ml | $13.78_{\pm 7.48}$ | $29.06_{\pm 9.31}$ |
| mr | $16.59_{\pm 9.30}$ | $33.66_{\pm 9.82}$ |
| gu | $18.21_{\pm 9.94}$ | $36.91_{\pm 11.55}$ |

Table 5: Average length of masked contexts per language in Camellia.

**Country distribution of Western entities.** Figure 7 reports the country-wise distribution of Western entities in `Camellia`. The countries of origin for authors, beverage, food, locations, and sport clubs, and entities were obtained from Wikidata which provides country of origin label for most entities, with the exception of some food and beverage entities that we manually annotated for origin. For Western first names, we prompted GPT-4o to classify the country of origin of each name to obtain the distribution in that entity type. We then manually verified these labels to be accurate. Examples include: Panagiotis as Greek, Javienne as French, Marilo as Italian, Erling as Norwegian, etc. We note that these country labels are visualization purposes and not used in our experiments.

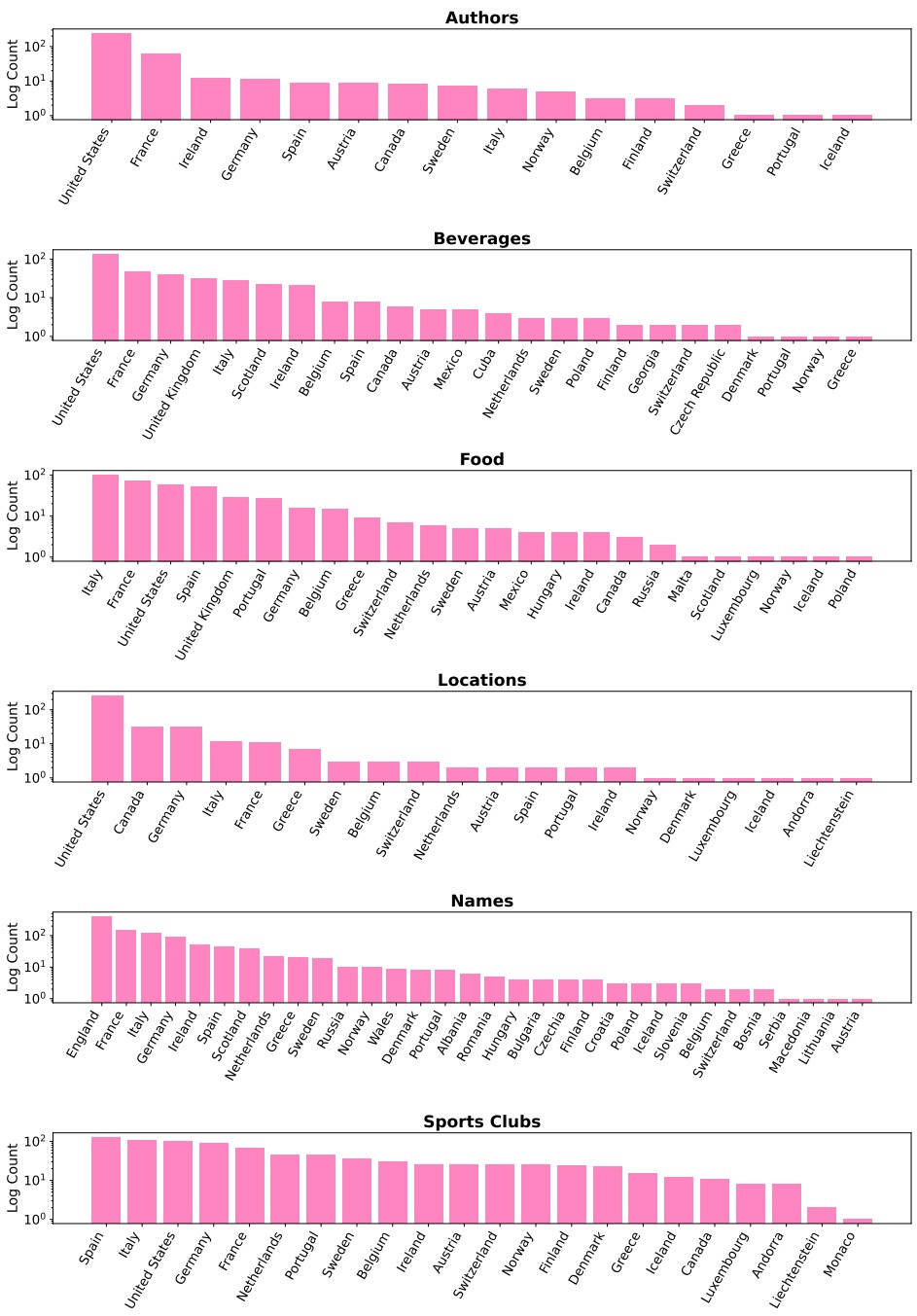

Figure 7: Country-wise distribution of Western entities in `Camellia` for different entity types.

| Language | Family | Morphology | Script |
|---|---|---|---|
| zh | Sino-Tibetan | Isolating | Logographic |
| ja | Japonic | Agglutinative | Logographic & Syllabic |
| ko | Koreanic | Agglutinative | Alphabetic (Hangul) |
| vi | Austroasiatic | Analytic/Isolating | Latin |
| ur | Indo-Aryan | Fusional | Perso-Arabic Nastaliq |
| hi | Indo-Aryan | Fusional | Alphasyllabary (Devanagari) |
| ml | Dravidian | Agglutinative | Alphasyllabary (Malayalam) |
| mr | Indo-Aryan | Fusional | Alphasyllabary (Devanagari) |
| gu | Indo-Aryan | Fusional | Alphasyllabary (Gujarati) |

Table 6: Typological Diversity of the languages in Camellia.

**Typological Diversity.** The languages in Camellia represent a broad span of typological diversity in terms of genealogical families, writing systems, and morphological profiles. We summarize those in Table 6 and report some details of each language below:

- **Chinese:** Chinese is a Sino-Tibetan language with a highly isolating morphology. It uses a logographic writing system (Han characters) that primarily encodes morphemes but also incorporates phonetic components, making it distinct from alphabetic scripts in terms of structure. The language is also tonal, adding further phonological complexity.

- **Japanese**: Japanese belongs to the Japonic family and exhibits agglutinative morphology (i.e, grammatical markers attach transparently to stems). Its writing system is tri-scriptal, combining Kanji (logographic) with Hiragana and Katakana (syllabaries).

- **Korean:** Korean is a Koreanic language. It uses Hangul, a featural alphabet whose letters combine into block-like syllabic units, creating a script that is alphabetic in design but syllabic in appearance. Korean is also agglutinative, with rich postpositional case particles and verbal morphology.

- **Vietnamese:** Vietnamese is an Austroasiatic language that is heavily shaped by historical Chinese contact. It has an analytic/isolating morphology with little inflection, and is tonal, distinguishing meaning through pitch contours. Its modern writing system is Latin-based but employs extensive diacritics for tones and vowel quality.

- **Urdu:** Urdu is an Indo-Aryan language with fusional morphology, expressing multiple grammatical categories through single affixes. It is written in Perso-Arabic script, a right-to-left script with complex ligatures and highly variable glyph shapes.

- **Hindi:** Hindi is an Indo-Aryan language that shares a lot of its grammatical structure with Urdu but differs in script. It has a fusional morphology, with rich agreement and case marking. Hindi uses the Devanagari alphasyllabary.

- **Malayalam:** Malayalam is a Dravidian language characterized by agglutinative morphology and long, morphologically complex word forms. Its Malayalam alphasyllabary has a large inventory of characters and ligatures.

- **Marathi:** Marathi is an Indo-Aryan language with fusional morphology and extensive nominal and verbal inflection. It is written in Devanagari but includes additional letters not found in Hindi, leading to differences in sound and usage.

- **Gujarati:** Gujarati is an Indo-Aryan language written in its own Gujarati alphasyllabary, which is historically related to but visually distinct from Devanagari. It exhibits fusional morphology with case marking, gender agreement, and verb inflection.

It is interesting to note that among the languages we study, four are gendered: Urdu, Hindi, Marathi, and Gujarati.

**Annotation Guideline**   Figure 8 shows our guideline for annotating cultural entities across all entity types, focusing on Indian culture for Hindi, Malayalam, Marathi, and Gujarati. We similarly adapted the guideline for the other cultures/languages by switching examples where necessary.

---

**Guideline for annotating entities for cultural association**

**(Hindi, Malayalam, Marathi, and Gujarati version)**

**Food entities:**

Classify the extraction according to the following labels:

- *Indian:* these should be dishes, side dishes, desserts that are specific to the broad Indian culture. For example, the dish "dosa" should be labeled as an Indian food entity. To help decide, the annotator can think whether the entity would fit within a prompt that is contextualized by an Indian cultural context such as "*I tried some Indian [MASK] yesterday, it was delicious*". These should be dishes originally from India.

- *Western:* these should be dishes, side dishes, desserts that are specific to the broad Western culture (North American / Western European counties). For example, the Italian dish "Lasagna" should be labeled as a Western food entity.

- *Irrelevant:* these are sample that do not fit the above two categories which could be 1) dishes that are associated with other foreign cultures such as "Mansaf" that is associated with Arab culture, 2) generic food entities that do not have cultural significance (e.g., bread, butter, olives, etc.), ingredients (e.g., cinnamon, saffron, etc.) or brands (cheetos, kinder, etc.) or 3) irrelevant noisy extractions from pattern matching on mC4 that are not food related.

**Beverage entities:**

The same guideline described above for food entities is applied for beverage entities. Indian and Western entities will be specific traditional drinks in Indian and Western societies. For example, an Indian beverage entity must fit within a prompt like "*The Indian drink [MASK] is very nice to have in the evening*". Examples of non-culture specific are "milk, tea, coca-cola", etc.

**Name entities:**

Name entities should be annotated as either "Indian" (e.g., Suraj, Naisha, etc.) or "Western" (e.g., Michael, Jessica, etc.). Filter out name entities that are neither Indian or Western such as names that are be associated with other foreign cultures (e.g., Arab, African, etc.) or irrelevant noisy extraction from pattern matching.

**Location / Authors / Sports Clubs entities:**

For these samples that are obtained from Wikidata using the country of origin tag, manually filter the entries to remove noisy samples from the database that are not associated with Indian culture (i.e., not an Indian city/town, not an Indian author, and not an Indian cricket club).

---

Figure 8: Indian-focused version of our annotation guideline for annotating cultural entities.

**Examples of Culturally-Grounded Contexts.** Figure 9 shows examples of culturally-grounded masked contexts for Chinese culture from Camellia-Grounded. In these examples, only entities associated with Chinese culture would be appropriate to fit the [MASK].

---

**Example culturally-grounded contexts for Chinese:**

**Authors:**

在中国，被称为文学家与革命家的完美结合的代表人物是[MASK]。
*Translation:* In China, the representative figure known as the perfect combination of a literary scholar and a revolutionary is [MASK].

**Beverage:**
在中国茶里，似江南佳人，凭淡雅茶香，令无数爱茶人迷醉的是[MASK]。
*Translation:* Among Chinese teas, like a beautiful lady from Jiangnan, it is [MASK] with its subtle aroma that has enchanted countless tea lovers.

**Food:**
闻起来臭，吃起来香，这就是来自中国长沙的经典美食[MASK]。
*Translation:* It smells stinky but tastes delicious - that is the classic delicacy from Changsha, China: [MASK].

**Locations:**
雪山、草甸、湖泊共同勾勒出如画美景，在中国川西路线上宛如仙境的地点是[MASK]。
*Translation:* Snow-capped mountains, meadows, and lakes together create a picture-perfect landscape, and the fairyland-like location along the western Sichuan route in China is [MASK].

**Names:**
昨天那场NBA比赛中国知名篮球解说员[MASK]对其进行了点评。
*Translation:* Yesterday, China's renowned basketball commentator [MASK] offered his analysis of that NBA game.

**Sports:**
在CBA的赛场上，辽宁男篮在客场以微弱优势战胜[MASK]，收获两连胜。
*Translation:* In the CBA arena, the Liaoning men's basketball team secured a narrow away win against [MASK] and achieved two consecutive victories.

在中超赛场上防守坚韧、进攻犀利，一路过关斩将，捍卫中国齐鲁足球荣耀的球队是[MASK]。
*Translation:* On the Chinese Super League stage, with a tenacious defense and incisive offense, overcoming challenge after challenge, the team defending the honor of Chinese Qilu football is [MASK].

---

Figure 9: Examples of culturally-grounded masked contexts for Chinese culture from Camellia-Grounded.

**Examples of Culturally-Neutral Contexts.** Figure 10 shows examples of culturally-neutral masked contexts for Chinese culture from Camellia-Neutral. In these examples, entities associated with any culture would be appropriate to fit the [MASK].

---

**Example culturally-grounded contexts for Chinese:**

**Authors:**

在中国，被称为文学家与革命家的完美结合的代表人物是*[MASK]*。
*Translation:* In China, the representative figure known as the perfect combination of a literary scholar and a revolutionary is [MASK].

**Beverage:**
在中国茶里，似江南佳人，凭淡雅茶香，令无数爱茶人迷醉的是[MASK]。
*Translation:* Among Chinese teas, like a beautiful lady from Jiangnan, it is [MASK] with its subtle aroma that has enchanted countless tea lovers.

**Food:**
闻起来臭，吃起来香，这就是来自中国长沙的经典美食[MASK]。
*Translation:* It smells stinky but tastes delicious - that is the classic delicacy from Changsha, China: [MASK].

**Locations:**
雪山、草甸、湖泊共同勾勒出如画美景，在中国川西路线上宛如仙境的地点是[MASK] 。
*Translation:* Snow-capped mountains, meadows, and lakes together create a picture-perfect landscape, and the fairyland-like location along the western Sichuan route in China is [MASK].

**Names:**
昨天那场NBA比赛中国知名篮球解说员[MASK]对其进行了点评。
*Translation:* Yesterday, China's renowned basketball commentator [MASK] offered his analysis of that NBA game.

**Sports:**
在CBA的赛场上，辽宁男篮在客场以微弱优势战胜[MASK]，收获两连胜。
*Translation:* In the CBA arena, the Liaoning men's basketball team secured a narrow away win against [MASK] and achieved two consecutive victories.

在中超赛场上防守坚韧、进攻犀利，一路过关斩将，捍卫中国齐鲁足球荣耀的球队是[MASK] 。
*Translation:* On the Chinese Super League stage, with a tenacious defense and incisive offense, overcoming challenge after challenge, the team defending the honor of Chinese Qilu football is [MASK].

---

Figure 10: Examples of culturally-neutral masked contexts for Chinese culture from Camellia-Neutral.

```
Classify the sentiment in this {LANGUAGE} sentence based on the following key:
0 = neutral
1 = positive
2 = negative

Sentence: ''{SENTENCE}''
Given the above key, the sentiment of this sentence is (0-2):
```

Table 8: Prompt used to classify a sentence's sentiment in our sentiment association experiment.

```
Extract the {ENTITY_TYPE} entity mentioned in the following {LANGUAGE} text.

Text: ''{QA_CONTEXT}''

Reply only with the mentioned {ENTITY_TYPE}. If nothing is found, reply ''None''.
```

Table 9: Prompt used to extract entities from contexts in our extractive QA experiment.

## B   ADDITIONAL EXPERIMENTAL DETAILS

**Prompts for extractive QA and sentiment classification.**   We used the same prompt used by Naous et al. (2024) for our sentiment association experiment, where models are given a key and asked to classify the sentiment of the given sentence (see Table 8). We also used the prompt by Naous & Xu (2025) for the extractive QA experiment, where models are given the context and entity type we seek to extract asked to identify the entity mentioned in the text (see Table 9).

**Inference Details and Parameters.**   We ran our experiments using 8 NVIDIA A40 GPUs. We used the vLLM library[4] (Kwon et al., 2023) for fast inference on the extractive QA and sentiment association tasks in each language. Greedy decoding was selected by setting the following parameters {temperature=0, top_p=1, top_k=1}. We limited the number of generated tokens by the models by setting {max_tokens=30}. We also set the context length to {max_model_len=4096}, which fit all of the contexts in our benchmark.

**Language Models.**   Table 7 lists the LLMs used in our experiments with their HuggingFace repositories. We used the largest size available for each LLM family and included the most recent version that mentions multilingual support (Llama, Qwen, Aya-expanse, and Gemma), and that we find to perform well enough on our tasks. However, we note that not all of these models we tested were explicitly developed to support the languages we evaluate on in Camellia. We also discarded certain recent models that we found not to perform well enough on some

| LLM | Hugging Face Repository |
|---|---|
| Llama3.3-70b | meta-llama/Llama-3.3-70B-Instruct |
| Qwen2.5-72b | Qwen/Qwen2.5-72B-Instruct |
| Aya-expanse-32b | CohereForAI/aya-expanse-32b |
| Gemma3-27b | google/gemma-3-27b-it |
| Olmo2-32b | allenai/OLMo-2-0325-32B-Instruct |
| Phi4-14b | microsoft/phi-4 |

Table 7: List of LLMs used with their Hugging Face repository links.

of our languages. We also use the Olmo2-32b and Phi4 models that are developed to handle English only for our English-only experiments. We also restricted our experiments to open-sourced models since we can obtain their log-probabilities, which are essential to compute the CBS scores in our cultural context adaptation experiment (§4.1).

---

[4]https://docs.vllm.ai

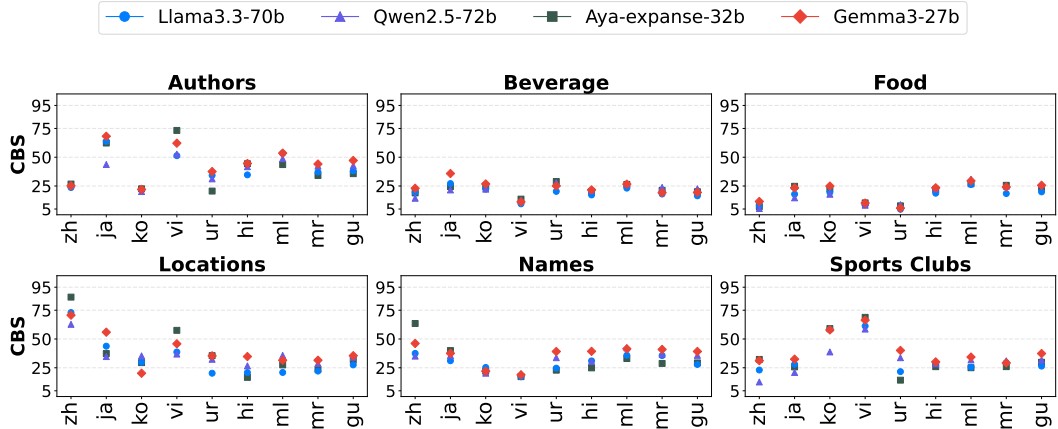

Figure 11: **C**ultural **B**ias **S**core (CBS) (↓) (§4.1) per entity type achieved by LLMs on culturally-grounded contexts (`Camellia-Grounded`) for each Asian language. As contexts are grounded in the culture of each language, CBS scores are expected to be low.

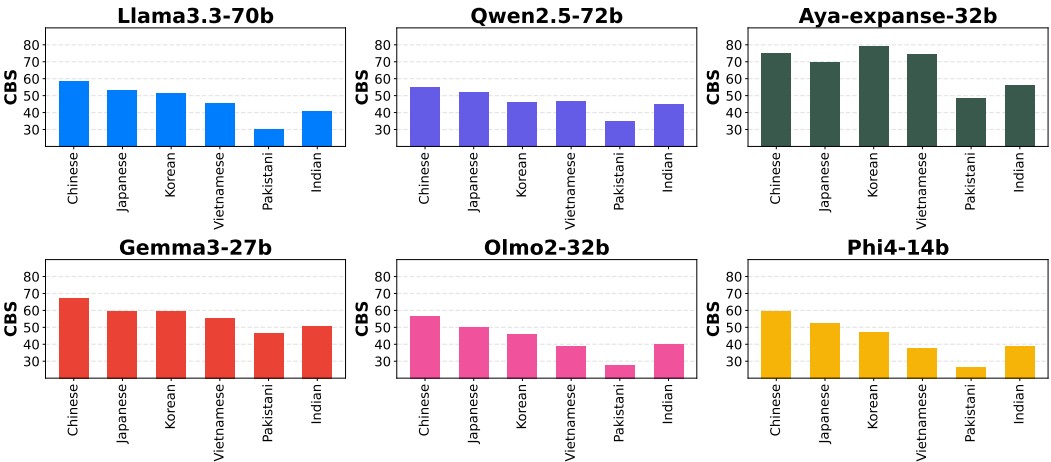

Figure 12: Average **C**ultural **B**ias **S**core (CBS) (↓) across entity types achieved by LLMs on culturally-grounded contexts (`Camellia-Grounded`) when tested in English for each culture.

## C  ADDITIONAL RESULTS

### C.1  CULTURAL ADAPTATION

**CBS scores per Entity Type.**  Figure 11 shows the CBS per entity-type achieved by LLMs when tested on the culturally-grounded contexts. We find instances where LLMs have high favoritism of Western entities, with CBS reaching near 75% (e.g., authors in `vi` and `ja`). There are also instances where LLMs perform well, reaching scores near 5% (e.g., food entities in `zh`, and `ur`).

**CBS scores when testing in English.**  Figure 12 shows the average CBS achieved by each model on the culturally-grounded contexts in `Camellia` when tested on the English translations for each culture. Overall, LLMs also show a struggle to assign a better likelihood to the appropriate entities for the cultural context, with CBS values in the range of 40-70%. The larger models (Llama3.3-70b and Qwen2.5-72b) perform better than smaller-sized models (Aya-expanse-32b and Gemma3-27b), suggesting that scaling can improve performance on this task. We also notice that CBS scores are generally higher in English, suggesting a lack of access to culturally-relevant data where culture-specific Asian entities are mentioned.

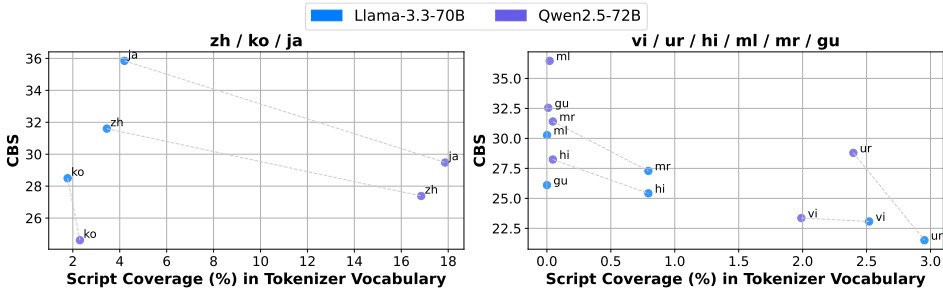

Figure 13: CBS vs script coverage % in tokenizer vocabulary for Llama3.3-70b and Qwen2.5-72b. Higher script coverage in a tokenizer tends to yield better performance (i.e., lower CBS). Dashed gray lines are shown between the results of both models for the same language for visual clarity. We note that both models had little to no coverage of the scripts for ml and gu.

**Tokenization Analysis.** The languages we study in Camellia span a wide range of writing systems. Chinese is written using a logographic script. Japanese combines logographic characters (Kanji) with two syllabaries (Hiragana and Katakana). Korean uses Hangul, an alphabet arranged into block-like characters. In contrast, the remaining languages use alphabetic systems, including Perso-Arabic script for Urdu, Brahmic scripts for Indian languages. The way these langauges are tokenized varies from one model to another.

To study the impact of tokenization differences across different models, we analyze the relationship between model performance and the coverage of each language's script within the tokenizer vocabulary. Specifically, for each model, we compute the percentage of tokens in its vocabulary containing at least one character from the script. We identify script-specific characters using their Unicode ranges (e.g., \u4E00-\u9FFF for Chinese, \u1100–\u11FF for Hangul, etc.). For Vietnamese, which uses the Latin alphabet with diacritical marks, we specifically count tokens containing such Vietnamese-specific markers (e.g., ă, â, ê, ô, à, á, , ã, etc.), ensuring we reflect tokens containing Vietnamese-specific characters rather than generic Latin script.

Figure 13 presents the CBS results for Llama-3.3-70B and Qwen-2.5-72B on all languages, plotted against each model's tokenizer script coverage. Overall, we observe that higher script coverage in a tokenizer tends to yield better performance (i.e., lower CBS). This trend is especially clear for Chinese, Japanese, and Korean, where Qwen outperforms Llama, consistent with Qwen's stronger coverage of these scripts. In contrast, for Hindi, Marathi, Urdu, and Vietnamese, the pattern reverses: Llama performs better, reflecting its better coverage of the scripts of these languages. As noted in prior studies (Foroutan et al., 2025), tokenization algorithms such as BBPE are trained on corpora with imbalanced language and script representation, which can place languages with underrepresented scripts at a disadvantage.

## C.2 SENTIMENT ASSOCIATION

**Test Set Sizes.** Table 10 reports the exact size of the test sets used in our sentiment association experiment (§ 4.2). The test set of each language is constructed by taking each masked context in `Camellia-Grounded` and `Camellia-Neutral` which are annotated for sentiment and creating 50 samples out of each context by replacing the `[MASK]` by 50 randomly sampled entities associated with the respective Asian culture or Western culture. Thus, the size of the Asian and Western test sets for each language is the same. We obtain test sets that range from generally range from 13,000 to 24,000 samples, depending on the amount of masked contexts we obtained in each language during data collection. We note that for Urdu the size of the test sets are smaller (2,550 samples each for Pakistani and Western) due to the language's low-resource nature and the limited availability of masked contexts.

| Language | Test Set Size |
|---|---|
| zh | 17,900 |
| ja | 13,850 |
| ko | 24,500 |
| vi | 17,550 |
| ur | 2,550 |
| hi | 19,882 |
| ml | 19,882 |
| mr | 19,882 |
| gu | 19,882 |

Table 10: Size of the native Asian and Western test sets used in our sentiment association experiment for each language.

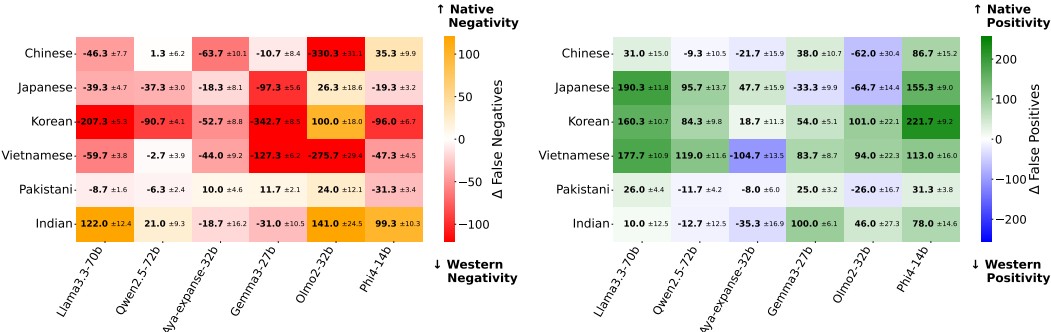

Figure 14: Differences in False Negative (FN) and False Positive (FP) sentiment predictions by LLMs on `Camellia` contexts filled with Asian vs Western entities, when tested in English. Results are averaged across 3 runs of 50 randomly sampled Asian vs Western entities in each culture.

**Results when testing in English.** Figure 14 shows the result of our sentiment association experiment when testing LLMs on the parallel English translations of the entities and contexts in each culture. In certain cases, the behavior of some models such as Gemma in English is consistent to when we tested in Asian languages, with generally more Western negativity and more positivity towards native Asian entities of each culture. There are certain cases where trends from the same model become different, such as for the Llama model, where it becomes more positive with native Asian entities in English.

## C.3 EXTRACTIVE QA

**Test Set Sizes.** Table 11 reports the exact size of the test sets used in our entity extractive QA experiment (§ 4.3). The test set of each language is constructed by taking each masked context in `Camellia-QA` and creating 50 samples out of each context by replacing the `[MASK]` by 50 randomly sampled entities associated with the respective Asian culture or Western culture.

| Language | Test Set Size |
|---|---|
| zh | 3,200 |
| ja | 3,000 |
| ko | 3,500 |
| vi | 3,900 |
| ur | 2,900 |
| hi | 2,350 |
| ml | 2,350 |
| mr | 2,350 |
| gu | 2,350 |

Table 11: Size of the native Asian and Western test sets used in our extractive QA experiment.

**Detailed Extractive QA Results.** Tables 12 and 13 show the detailed accuracy results on the extractive QA task. We compute accuracy based on the exact match of identifying the entity in the context. We observe large accuracy gaps between sets containing Asian and Western entities when testing in the respective Asian language of each culture, where LLMs mostly perform better at extracting Asian-associated entities. In contrast, these gaps are negligible in English in nearly all cases (2%-5% gaps). In a couple of cases, large gaps in English are observed (Pakistani vs Western entities in Llama and Qwen, Indian vs Western entities in Qwen).

| | Llama3.3-70b | | | | | | Qwen2.5-72b | | | | | |
|---|---|---|---|---|---|---|---|---|---|---|---|---|
| Test Lang | **Respective Asian** | | | **English** | | | **Respective Asian** | | | **English** | | |
| Culture | *Asian* | *Western* | ΔAcc | *Asian* | *Western* | ΔAcc | *Asian* | *Western* | ΔAcc | *Asian* | *Western* | ΔAcc |
| Chinese | 94.81 | 96.13 | -1.32 | 91.42 | 91.11 | 0.30 | 95.46 | 95.03 | 0.43 | 88.57 | 91.41 | -2.84 |
| Japanese | 91.49 | 83.94 | 7.55 | 92.48 | 89.77 | 2.72 | 88.47 | 69.60 | 18.87 | 88.44 | 83.90 | 4.53 |
| Korean | 91.74 | 82.06 | 9.69 | 92.34 | 91.69 | 0.66 | 91.17 | 74.70 | 16.47 | 85.14 | 87.63 | -2.49 |
| Vietnamese | 74.78 | 88.31 | -13.53 | 91.70 | 89.75 | 1.95 | 73.67 | 88.00 | -14.33 | 83.44 | 87.05 | -3.61 |
| Pakistani | 75.42 | 80.13 | -4.71 | 99.66 | 89.11 | 10.54 | 67.73 | 72.71 | -4.99 | 99.77 | 87.61 | 12.16 |
| Indian (hi) | 95.45 | 85.40 | 10.05 | 98.31 | 91.59 | 6.71 | 70.38 | 66.74 | 3.63 | 98.06 | 87.38 | 10.67 |
| Indian (ml) | 76.09 | 62.94 | 13.15 | — | — | — | 55.73 | 51.51 | 4.22 | — | — | — |
| Indian (mr) | 94.45 | 83.38 | 11.07 | — | — | — | 48.58 | 46.90 | 1.68 | — | — | — |
| Indian (gu) | 87.56 | 73.12 | 14.44 | — | — | — | 50.43 | 44.40 | 6.02 | — | — | — |

Table 12: Detailed accuracy results for Llama3.3-70b and Qwen2.5-72b on the extractive QA task when tested in the respective Asian language of each culture vs. in English.

| | Aya-expanse-32b | | | | | | Gemma3-27b | | | | | |
|---|---|---|---|---|---|---|---|---|---|---|---|---|
| Test Lang | **Respective Asian** | | | **English** | | | **Respective Asian** | | | **English** | | |
| Culture | *Asian* | *Western* | ΔAcc | *Asian* | *Western* | ΔAcc | *Asian* | *Western* | ΔAcc | *Asian* | *Western* | ΔAcc |
| Chinese | 87.08 | 84.24 | 2.84 | 81.08 | 86.91 | -5.83 | 91.58 | 92.94 | -1.36 | 84.13 | 89.76 | -5.63 |
| Japanese | 86.96 | 78.12 | 8.84 | 83.77 | 84.51 | -0.73 | 81.84 | 65.44 | 16.40 | 83.97 | 87.19 | -3.22 |
| Korean | 93.20 | 79.26 | 13.94 | 95.51 | 94.09 | 1.43 | 92.43 | 84.49 | 7.94 | 96.71 | 94.17 | 2.54 |
| Vietnamese | 76.56 | 73.73 | 2.83 | 91.09 | 92.97 | -1.88 | 93.87 | 89.72 | 4.15 | 97.66 | 96.01 | 1.65 |
| Pakistani | 66.66 | 66.53 | 0.12 | 97.61 | 93.08 | 4.54 | 81.75 | 60.64 | 21.11 | 99.53 | 95.00 | 4.54 |
| Indian (hi) | 86.39 | 74.85 | 11.54 | 94.62 | 93.55 | 1.07 | 85.72 | 78.91 | 6.81 | 98.52 | 95.26 | 3.25 |
| Indian (ml) | 70.46 | 59.52 | 10.93 | — | — | — | 52.87 | 43.85 | 9.01 | — | — | — |
| Indian (mr) | 81.84 | 69.20 | 12.64 | — | — | — | 86.80 | 83.29 | 3.50 | — | — | — |
| Indian (gu) | 65.19 | 52.30 | 12.89 | — | — | — | 86.30 | 79.76 | 6.54 | — | — | — |

Table 13: Detailed accuracy results for Aya-expanse-32b and Gemma3-27b on the extractive QA task when tested in the respective Asian language of each culture vs. in English.

## C.4 Leaderboards

We report leaderboards for our of our tasks where we average the results across all languages.

**Multilingual Leaderboards.** We report leaderboards when testing on all the Asian languages in Camellia: Table 14 for cultural adaptation, Tables 15 and 16 for sentiment association, and Table 17 for entity extractive QA.

| Model | Cultural Bias Score (CBS) (↓) | | | | | | | | | |
|---|---|---|---|---|---|---|---|---|---|---|
| | zh | ja | ko | vi | ur | hi | ml | mr | gu | Avg (↓) |
| Llama3.3-70b | 31.74 | 34.68 | 28.83 | **29.11** | **21.51** | 26.03 | 31.56 | 28.06 | **26.31** | **28.65** |
| Aya23-expanse-32b | 42.64 | 36.37 | 28.96 | 36.88 | 21.84 | **25.60** | **30.79** | 26.88 | 28.49 | 30.94 |
| Qwen2.5-72b | **27.36** | **29.56** | **24.99** | 29.78 | 28.79 | 28.76 | 36.91 | 31.98 | 32.72 | 30.10 |
| Gemma3-27b | 36.27 | 41.55 | 27.99 | 33.34 | 31.73 | 33.28 | 36.92 | 32.86 | 34.77 | 34.30 |

Table 14: Leaderboard for our cultural adaptation experiment. Lower CBS reflects better adaptation.

| Model | \|ΔFalse Negatives\| (↓) | | | | | | | | | |
|---|---|---|---|---|---|---|---|---|---|---|
| | zh | ja | ko | vi | ur | hi | ml | mr | gu | Avg (↓) |
| Aya23-expanse-32b | **19.0** | 49.0 | **11.67** | 34.67 | 18.0 | **10.0** | 36.33 | **16.33** | **16.67** | **23.52** |
| Qwen2.5-72b | 43.67 | **2.67** | 26.0 | 30.33 | **1.0** | 65.33 | **22.67** | 41.0 | 69.33 | 33.56 |
| Llama3.3-70b | 45.67 | 28.33 | 85.33 | **22.67** | 2.33 | 88.67 | 115.33 | 43.67 | 77.67 | 56.63 |
| Gemma3-27b | 72.33 | 74.33 | 226.33 | 115.0 | 12.67 | 131.0 | 34.0 | 61.67 | 103.67 | 92.33 |

Table 15: Leaderboard for our negative sentiment association experiment. A lower |ΔFalse Negatives| reflects lower predictions of false negative sentiment associations.

| Model | \|ΔFalse Positives\| (↓) | | | | | | | | | |
|---|---|---|---|---|---|---|---|---|---|---|
| | zh | ja | ko | vi | ur | hi | ml | mr | gu | Avg (↓) |
| Llama3.3-70b | 73.33 | 49.33 | 20.33 | 90.67 | 6.0 | 33.0 | 32.33 | 84.67 | 34.33 | **47.11** |
| Gemma3-27b | 33.67 | 11.33 | 255.67 | 11.67 | 2.33 | 10.67 | 110.67 | 27.0 | 6.67 | 52.19 |
| Aya23-expanse-32b | 58.67 | 81.67 | 170.0 | 72.33 | 25.33 | 3.67 | 4.0 | 131.0 | 108.0 | 72.74 |
| Qwen2.5-72b | 55.67 | 19.67 | 79.33 | 32.67 | 10.0 | 134.0 | 250.33 | 187.0 | 189.0 | 106.41 |

Table 16: Leaderboard for our positive sentiment association experiment. A lower |ΔFalse Positives| reflects lower predictions of false positive sentiment associations.

| Model | \|ΔQA Accuracy\| (↓) | | | | | | | | | |
|---|---|---|---|---|---|---|---|---|---|---|
| | zh | ja | ko | vi | ur | hi | ml | mr | gu | Avg (↓) |
| Qwen2.5-72b | **0.43** | 18.87 | 16.47 | 14.33 | 4.99 | **3.63** | **4.22** | **1.68** | 6.02 | **7.85** |
| Aya23-expanse-32b | 2.84 | 8.84 | 13.94 | **2.83** | **0.12** | 11.54 | 10.93 | 12.64 | 12.89 | 8.51 |
| Gemma3-27b | 1.36 | 16.40 | **7.94** | 4.15 | 21.11 | 6.81 | 9.01 | 3.50 | 6.54 | 8.54 |
| Llama3.3-70b | 1.32 | **7.55** | 9.69 | 13.53 | 4.71 | 10.05 | 13.15 | 11.07 | 14.44 | 9.50 |

Table 17: Leaderboard for our extractive QA experiment. Lower |ΔQA Accuracy| reflects less performance gaps in entity extractions between the native Asian culture and Western culture.

**English Leaderboards.** We report leaderboards when testing on all the Asian cultures in Camellia in the English language: Table 18 for cultural adaptation, Tables 19 and 20 for sentiment association, and Table 21 for entity extractive QA.

| Model | Cultural Bias Score (CBS) (↓) | | | | | | |
|---|---|---|---|---|---|---|---|
| | Chinese | Japanese | Korean | Vietnamese | Pakistani | Indian | **Avg** (↓) |
| Olmo2-32b | **56.39** | **50.01** | **45.83** | 38.92 | 27.33 | 39.76 | **43.04** |
| Phi4-14b | 59.51 | 52.16 | 47.10 | **37.62** | **26.24** | **38.94** | 43.59 |
| Qwen2.5-72b | 55.15 | 51.91 | 46.25 | 46.43 | 34.64 | 44.93 | 46.55 |
| Llama3.3-70b | 58.45 | 53.47 | 51.68 | 45.35 | 30.36 | 41.05 | 46.73 |
| Gemma3-27b | 67.10 | 59.10 | 59.18 | 55.42 | 46.46 | 50.35 | 56.27 |
| Aya23-expanse-32b | 75.35 | 70.01 | 79.35 | 74.19 | 48.55 | 55.97 | 67.24 |

Table 18: English Leaderboard for our cultural adaptation experiment. Lower CBS reflects better adaptation.

| Model | $|\Delta$False Negatives$|$ (↓) | | | | | | |
|---|---|---|---|---|---|---|---|
| | Chinese | Japanese | Korean | Vietnamese | Pakistani | Indian | **Avg** (↓) |
| Qwen2.5-72b | **1.33** | 37.33 | 90.67 | **2.67** | **6.33** | 21.0 | **26.56** |
| Aya23-expanse-32b | 63.67 | **18.33** | **52.67** | 44.0 | 10.0 | **18.67** | 34.56 |
| Phi4-14b | 35.33 | 19.33 | 96.0 | 47.33 | 31.33 | 99.33 | 54.78 |
| Llama3.3-70b | 46.33 | 39.33 | 207.33 | 59.67 | 8.67 | 122.0 | 80.56 |
| Gemma3-27b | 10.67 | 97.33 | 342.67 | 127.33 | 11.67 | 31.0 | 103.44 |
| Olmo2-32b | 330.33 | 26.33 | 100.0 | 275.67 | 24.0 | 141.0 | 149.56 |

Table 19: English leaderboard for our negative sentiment association experiment. A lower $|\Delta$False Negatives$|$ reflects lower predictions of false negative sentiment associations.

| Model | $|\Delta$False Positives$|$ (↓) | | | | | | |
|---|---|---|---|---|---|---|---|
| | Chinese | Japanese | Korean | Vietnamese | Pakistani | Indian | **Avg** (↓) |
| Aya23-expanse-32b | 21.67 | 47.67 | **18.67** | 104.67 | **8.0** | 35.33 | **39.33** |
| Qwen2.5-72b | **9.33** | 95.67 | 84.33 | 119.0 | 11.67 | **12.67** | 55.44 |
| Gemma3-27b | 38.0 | **33.33** | 54.0 | **83.67** | 25.0 | 100.0 | 55.67 |
| Olmo2-32b | 62.0 | 64.67 | 101.0 | 94.0 | 26.0 | 46.0 | 65.61 |
| Llama3.3-70b | 31.0 | 190.33 | 160.33 | 177.67 | 26.0 | 10.0 | 99.22 |
| Phi4-14b | 86.67 | 155.33 | 221.67 | 113.0 | 31.33 | 78.0 | 114.33 |

Table 20: English leaderboard for our positive sentiment association experiment. A lower $|\Delta$False Positives$|$ reflects lower predictions of false positive sentiment associations.

| Model | $|\Delta$QA Accuracy$|$ (↓) | | | | | | |
|---|---|---|---|---|---|---|---|
| | Chinese | Japanese | Korean | Vietnamese | Urdu | Indian | **Avg** (↓) |
| Aya23-expanse-32b | 5.83 | **0.73** | 1.43 | 1.88 | **4.54** | 1.07 | **2.58** |
| Olmo2-32b | 4.36 | 0.37 | 1.46 | 3.10 | 4.50 | 1.85 | 2.61 |
| Gemma3-27b | 5.63 | 3.22 | 2.54 | **1.65** | 4.54 | 3.25 | 3.47 |
| Llama3.3-70b | **0.30** | 2.72 | **0.66** | 1.95 | 10.54 | 6.71 | 3.81 |
| Qwen2.5-72b | 2.84 | 4.53 | 2.49 | 3.61 | 12.16 | 10.67 | 6.05 |

Table 21: English leaderboard for our extractive QA experiment. Lower $|\Delta$QA Accuracy$|$ reflects less performance gaps in entity extractions between the native Asian culture and Western culture. We omit Phi4-14b due to its low performance on this task.