# OpenReview forum: "Camellia: Benchmarking Cultural Biases in LLMs for Asian Languages"
_ICLR.cc/2026/Conference — Submitted to ICLR 2026_

### Official Review · Reviewer_Cxnr · 2025-10-27

**Soundness:** 2
**Presentation:** 3
**Contribution:** 2
**Rating:** 4
**Confidence:** 4

**Summary:**

This paper introduces the Camellia benchmark for evaluating cultural bias surrounding entities from 9 Asian languages against a collection of ‘Western’ entities. Based on the concepts and approaches from [Naous et al. 2024], the benchmark consists of manually annotated entities from different categories that each come with masked context derived from Tweets. The benchmark consists of three tasks: Context adaption, sentiment association, and entity extraction. The paper then uses the benchmark to evaluate four different models showing biased model behavior across all three tasks

**Strengths:**

The paper looks at the interesting problem of quantifying entity bias of LLMs. Here, the paper identifies a set of understudied Asian cultures for which a benchmark is built, thus enabling the evaluation of entity bias w.r.t. these cultures.

The authors provide, at least partial, human annotation and validation, thus creating a more trustworthy resource than the often fully automatically generated benchmarks from related work.

The writing is clear and engaging.

**Weaknesses:**

**W1)** Methodologically, the paper is mostly taking the work in [Nous et al, 2024] and applying it to a new set of languages. Dataset creation and annotation is of course valuable work, especially when involving human annotation. However, this limits the methodological novelty of the work.

**(W2)** While defining and naming cultural groups is a challenging task, the authors often lack arguments for their choices and do not discuss them in conjunction with related work (where they provide nearly no reference for any of their choices). This results in several issues:

- The paper’s motivation revolves around the discrimination of an overgeneralized ‘Western culture’ group (line 013, based on the findings in  [Naous et al. 2024]). The authors' findings are more nuanced, however, showing better and worse performance for either region, depending on the setting. It is not clear, why this Asian-Western definition was chosen over, e.g., ‘high- vs. low-resource’ or ‘dataset frequency’.

- Similarly, why is the baseline group chosen as ‘Western’ and not, e.g., ‘U.S. American’ to compare at similar levels of granularity (i.e. country level)? It is likely that this experimental choice introduces bias that is not accounted for or discussed (e.g. low-resource languages in both Europe and Asia being underrepresented).

- The authors losely define "Western" as “North America and Europe” (line 103), missing possible other "Western"-dominated cultures like Australia and including Mexico, which is often more associated with Latin America, but on the North American continent. While arguments could be made for these choices (e.g. Mexico's cultural closeness to the Southern part of the US), the authors do not provide this argumentation.

- The notion of a benchmark for ‘Asian languages’ also implies a more diverse a selection of languages than the ones considered which are limited to 9 Asian languages from East and South Asia.

**(W3)** Given the complexity of the topic, the authors need to be careful how they phrase their statements. E.g. a bolded statement like "LLMs still lack a robust ability to grasp implicit contexts in most non-English languages" (line 373) implies that a broad set of non-English languages have been analyzed. However, the authors only study their subset of Asian languages in this experiment, missing most of the rest of the world (incl. Europe, South America, Africa and parts of Asia) and thus making the "non-English languages" statemtent problematic. Similarly, sentences like ”Our analyses show a struggle by LLMs at cultural adaptation in all Asian languages,” (line 022) can be prone to overinterpretatation of the results provided (assuming a broad coverage of Asian languages, which is not the case. I urge the authors to be careful in their phrasing, using e.g. "all studied Asian languages" instead and avoiding overgeneralizations like "non-English languages".

**(W3)** The paragraph starting from line 203 (“Contexts for Evaluating Cultural Adaption.”) is lacking a precise definition of what constitutes a ‘cultural context’ and what criteria were used to annotate them. This creates a challenge for interpreting the results.

**(W4)** Human verification is lacking for some dataset creation tasks and some tasks would need human quality control assessments.
- The authors use double annotation for one aspect (cultural origin of an entity), the authors do not perform this (or at least do not report it) for the other manual annotation steps. This is especially relevant for the sentences that are only appropriate in one cultural context vs. all contexts (Camelia-Grounded/Neutral) and for the sentiment task (line 209). Inter-annotator agreement would be important here to understand the quality and difficulty of these annotations, as these are at the core of the following experiments.
- Less crucial, but double annotation/inter-annotator agreement would also be interesting for other manual tasks like selection of culturally relevant entities, translations/transliterations into English or translations of Western languages into low-resource languages.
- The authors do not evaluate some of their automated steps (e.g. the automatic parallelization of Entities in Indian Languages via English).

**(W5)** Figure 2 shows that the entities from each ‘culture’ are not distributed equally. In 5/6 categories the ‘Western’ group has more entitites than the majority of the other ‘cultures’.

**(W6)** Section 2 reports a straightforward approach to creating the dataset for multiple languages. Section 4 then presents exceptions to this process (e.g. using governmental reports and name generators for Chinese, Korean and Japanese). While I very much appreciate that the authors consider these language nuances, this should have been part of the dataset creation description.

**Minor aspects**
- It would have been nice to include a short discussion in the related work about cultural representation in general and entities just being one dimension (cf. also https://aclanthology.org/2025.tacl-1.31/ ).
- Cultural evaluation is a rapidly growing area and the authors list a lot of relevant works. Fitting to their entity-focused approach, the following recent works could also be relevant: https://aclanthology.org/2024.acl-long.345/  https://arxiv.org/abs/2505.21693 https://aclanthology.org/2025.naacl-long.402/
- The authors report averages over several runs in Figure 5 but miss any form of variance.
- In Eq. 1, I would write the function that is defined before the equation, i.e. CBS_D … = Equation 1
- Would it make sense to report baseline values for the CBS from other studies instead of just stating 'the CBS is expected to be low’ in line 267? Without any substantiation this claim does not seem intuitive.

**Questions:**

1. Would it make sense to study and refer to secondary literature for the definition of the cultural categories of the paper? What would that change in your setting? (see W2)

2. How is cultural context defined? (see W3)

3. What are the results of a human verification and quality control assessment? (see W4)

4. Is the inbalance of the entities an issue? (see W5)

---

> ### Author Response · Authors · 2025-11-22
> **Response to Reviewer Cxnr 1/2**
>
> Thank you so much for your efforts reviewing our paper, we really appreciate it! We respond to your concerns and questions below.
>
> **(W1) Methodological Novelty:** The main contribution of our work is introducing the Camellia benchmark for measuring entity-centric cultural biases in 9 diverse Asian languages. Past resources that enable this type of evaluation were limited to only Arabic language, which makes analyses across different non-Western languages difficult. To address this, we manually collected and annotated 19,530 cultural entities extracted from Wikidata and mC4 web-crawls and 2,173 natural masked contexts constructed from social media posts in all 9 languages, which required a large amount of manual effort by native speakers. We believe Camellia will be a critical evaluation resource for addressing these entity-centric issues in future LLMs.
>
> **(W2) Definition of Western and Asian entities:** We believe that for natives of the Asian cultures we study, there is a clear distinction between entities associated with their native Asian culture and entities seen as “Western”. For example, a Chinese person would associate the first name “Weili” as being Chinese and the first name “Valentina” as being Western. We provide such examples from each Asian culture in Figure 2 and contrast them with Western entities. We grouped entities from Northern America (including Mexico) and Europe into a broad “Western culture” as viewed from the perspective of these Asian cultures. Since entities from the many different Western countries appear sparsely in these different Asian languages, grouping them into one Western group simplifies the design of our benchmark and the interpretation of our results. We agree that we do not include other Western-dominated areas such as Australia, but we would be happy to discuss this as a limitation.
>
> **(W3) Rephrasing Statements:** We have rephrased these statements in the paper to state the “studied Asian languages in Camellia” instead of “all Asian languages” to avoid overgeneralization. Thank you for pointing this out!
>
> **(W3) Definition of cultural context:** We clarify that by “cultural context” we refer to contexts in which the [MASK] is only suitable for an entity associated with that culture and does not fit entities from other cultures. For example, consider a masked context such as “I tried the Chinese dish [MASK]”. This is a cultural context for Chinese culture, where the [MASK] is only suitable to place a Chinese dish and not something from a different culture. We have modified our writing from line 203 to better describe our definition of cultural context and improve clarity!
>
> **(W4) Human Verification of Masked Contexts:** All of the masked contexts Camellia were carefully constructed by authors of the paper who are native speakers of each language. This was done by manually searching for discussions on social media and identifying tweets from which have a clear cultural grounding (as we define in our response to W3 above) and ones that are culturally neutral. We have included a big list of examples from the dataset of culturally-grounded and culturally-neutral contexts for Chinese culture in Appendix A (Figures 8 and 9). We hope the examples are sufficient to show the clear distinction between both types of contexts.
>
> **(W4) Validity of Translations:** We note that a lot of manual effort was done to ensure the quality of the translations. When parallelizing Western entities in Camellia, we took a random sample of 500 per entity type to keep the amount of translations at a reasonable amount for manual translation efforts and ensure high quality. To further ensure quality, we are performing a post-annotation quality check on random samples of 100 entities with external native speaker volunteers that were not involved in this study and have achieved the following substantial agreements (zh: 96%, ko: 95%, ja: 96%, vi: 97%, ur: 99%, ml: 99%, mr: 100%, gu: 99%). We hope this further shows the quality of our translations and we are happy to include this additional check in the paper if you believe it is helpful!
>
>
> **(W5) Imbalance of entities across cultures:** We clarify that the imbalance in the number of entities for different cultures is natural and is not an issue. Naturally, some cultures will have more entities than others. For example, China has a much larger population and geographic landscape, which results in more food and location entities.  For each culture, we did our best effort to collect and annotate an exhaustive number of entities (Wikidata + pattern-based extraction from mC4 webcrawls), from which we then take random samples in our experiments and report the average of results.
>
> **(W6) Section 4 content:**  We have incorporated your suggestion and moved the content of section 4 about language nuances into the dataset section for a better flow.

---

> > ### Author Response · Authors · 2025-11-22
> > **Response to Reviewer Cxnr 2/2**
> >
> > **(Minor) Related work:** We have included the suggested papers into our related work and moved this section to the earlier part of the paper after our introduction. In the first part of our related work, we provide an overview of other cultural evaluation resources that follow a question-answering format or cultural knowledge bases before referencing related papers focused on entities.
> >
> >
> > **(Minor) Variance for results:** We have included the variance of the results to our plots.
> >
> > **(Minor) Equation 1 format:** We have updated the format according to your suggestion (CBS = Equation 1).
> >
> > **(Minor) CBS clarification:** We have clarified this statement and referenced the paper that introduced the CBS for contextualization, thanks for this suggestion!
> >
> > The mentioned changes can be seen in the updated PDF of the submission. We hope that our response and modifications addressed your concerns and would really appreciate it if you could increase your score!

---

> ### Comment · Reviewer_Cxnr · 2025-11-26
>
> Thank you for adressing my concerns.
>
> (W1) I fully agree that creating such dataset resources is a lot of effort and that they can be very valuable to the community. My only argument there was that this limits the *methodological* novelty (which is of course not strictly necessary for a good paper).
>
> (W2) I understand your arguments for taking these decisions. It would be great if they are spelled out in the paper itself so that the reader is aware of the choices you have taken (transparency in research). Even the fact that you count Mexico as a Western country (but not other Latin American countries) is not mentioned in the paper itself, as far as I can see.
>
> Regarding the discussion about the limitation with Australia you proposed: Any reason you did not include that in the updated PDF?
>
> (W3-W6) Thanks for performing the additional verification and implementing the other changes.
>
> If you would add the transparent discussion of how you define Western and Asian cultural contexts (what you include, what you exclude, and why), I'd be happy to raise my score to 6.

---

> > ### Author Response · Authors · 2025-11-26
> >
> > Thanks a lot for your response!
> >
> > We have added a new discussion at the beginning of Section 3.1 that explains how we define and distinguish the Asian and Western cultures in our paper. We also mentioned the limitation regarding the exclusion of certain regions that are Western-dominated such as Australia, and included a limitations section at the end of the paper where we discuss this (after the conclusion section). For further transparency, we have also reported the country-wise distribution of the Western entities in Camellia in Appendix A (Figure 7). The PDF has been updated with these changes.
> >
> > Thank you so much again for your time and effort in providing valuable comments and feedback that helped improve the quality of our paper! We hope that we have addressed all of your concerns and would really appreciate it if you would increase your score to 6!

---

### Official Review · Reviewer_v7VU · 2025-10-27

**Soundness:** 3
**Presentation:** 4
**Contribution:** 3
**Rating:** 6
**Confidence:** 2

**Summary:**

This paper benchmarks cultural biases in LLMs with a focus on Asian languages and cultures, by evaluating models (four open multilingual model families) via context preference, sentiment association and extractive QA tasks.
It largely follows CAMel (Naous et al. 2024) that did similar work for Arabic and Arab culture.
Data is extracted from Wikidata, web crawls, and from X.
They find that most LLMs struggle with distinguishing between Asian and Western entities in Asian culture contexts (i.e. scoring Western entities highly). Trends of sentiment across models and cultures appear to diverge, there is high variance based on the individual models and languages/cultures. For context extraction, there is still headroom compared to English performance. The paper also provides a discussion on entity-based research in the multilingual space.

Overall a nice and valuable contribution on an important topic (closing gaps in multilingual LLM evaluation), however, in my opinion not novel enough in terms of the methodology for ICLR, and fairly shallow in relating findings to prior work. This might be better placed at an *ACL venue with an explicit focus on language expansion.

**Strengths:**

- Important topic: while most LLMs are multilingual, their evaluations are missing for a lot of languages and cultures. Expanding the language coverage of evaluations is helpful to understand where today's models are not sufficiently multilingual yet.
- Necessary care for detail on the data curation side.
- Well structured and written, with a particularly nice overview figure on the first page.

**Weaknesses:**

- Context of other works could be made more clear: Which findings align with related works on similar cultural benchmarking? These papers are listed in Related Work, but not really taken into account in any discussion.
- Lack of novelty in methodology: the paper largely folllows CAMel, and does not clearly indicate where an if it diverges or innovates. The paper refers to it in various places but it is not clear which aspects of the methodology or insights are new (beyond the obvious language/region expansion).
- Some open questions in experiment design, evaluation and data validation, see below.

**Questions:**

- Is using X as a source truly indicative of representative native speaker discussions? It seems like X is blocked in some countries (e.g. China, Iran, North Korea, Myanmar, and Russia). This includes relevant regions for this study (Asia). The data sourced from X is claimed to contain “natural discussions by native speakers” - how is this ensured/validated? The paper mentions manual inspection of retrieved tweets - how and by whom is this done?
- The data is web-sourced, so why are models that are trained on web sources not better?
- The language support is not considered in the evaluation. Some models explicitly state language support, which should be taken into account in the evaluation. E.g. Aya Expanse 32B does not support Urdu - the bias is low but probably the uncertainty is very high in general.
- How can the differences between the results of the different benchmarking formats be explained?
- How were data filters adapted to the languages of study? (Or made sure that they’re appropriate)
- How long are contexts?

**Details Of Ethics Concerns:**

Afaik crawling X is banned. It seems like the data extraction on X might violate this ban.

---

> ### Author Response · Authors · 2025-11-22
> **Reviewer v7VU**
>
> Thank you so much for your efforts reviewing our paper, we really appreciate it! We respond to your concerns and questions below.
>
> **(W1) Methodological Novelty:** The main contribution of our work is introducing the Camellia benchmark for measuring entity-centric cultural biases in 9 diverse Asian languages. Past resources that enable this type of evaluation were limited to only Arabic language, which makes analyses across different non-Western languages difficult. To address this, we manually collected and annotated 19,530 cultural entities extracted from Wikidata and mC4 web-crawls and 2,173 natural masked contexts constructed from social media posts in all 9 languages, which required a large amount of manual effort by native speakers. We believe Camellia will be a critical evaluation resource for addressing these entity-centric issues in future LLMs.
>
> **(W2) Results in context to other works**: We have moved our related work section to the earlier part of the paper for clearer contextualization of our paper with the literature. We also included relevant references when discussing the results on each of our tasks. For example, we point to the study of Guo et al. 2025 which has always found a similar trend in performance on cultural tasks when comparing model families developed in different regions. We also point to the studies of Li et al. 2025 and Naous et al. 2025 when discussing our results on the extractive QA task, which corroborate our findings regarding the issue of LLMs in handling long-tail entities and their struggle on multi-sense words that overlap with cultural entities.
>
> **(Q1) Data source and collection clarification:** It is true that X is banned in the countries you mentioned, and this only impacts Chinese in our dataset and not any of the other languages we study. We clarify that, differently to the other languages, we collected Chinese contexts from the Weibo and Xiaohongshu platforms which are popular in China. We apologize for not mentioning this Chinese-specific detail, we have now clarified it in the manuscript. The inspection of tweets/posts to find suitable contexts was done by the authors themselves, where an author that is a native speaker of each language performed this inspection to ensure the quality of the text. We have included a list of examples of the contexts from the dataset in Appendix A (Figures 8 and 9).
>
> **(Q2) Models trained on web sources:** Unfortunately, all of the recent multilingual LLMs do not open-source their training data, which limits the ability to perform further analysis on their pre-training data domains and how it impacts performance. However, it is known that all LLMs are trained on web sources in the form of web-crawled data such as CommonCrawl scrapes.
>
> **(Q3) Language Support of LLMs:** We made our best effort to include the most recent LLMs with multilingual support, which are unfortunately not many. We believe that the models we selected (Llama, Qwen, Gemma, Aya-expanse) achieved a sufficient level of performance for our tasks and thus reported their results. We also note that we did exclude several models which we found not to work well on all of the languages we study (such as GPT-oss). We have clarified this in our Appendix B.
>
> **(Q4) Result trends for different tasks:** This is an interesting point! We hypothesize that the trends shown by models on different tasks stem from different reasons. For example, the results on entity sentiment association are likely due to co-occurrences with negative or positive framings that LLMs are exposed to in the training data, which varies from one model to the other.. On the other hand, cases where the models show similar behavior such as extractive QA tasks may stem from other reasons such as linguistic phenomena that current LLMs struggle to handle, as pointed out by prior studies. We hope that Camellia will encourage further analyses into these reasons.
>
> **(Q5) Data filtering:** We clarify that all steps that required data filtering were done manually by authors that are native speakers for each language. We put a lot of manual effort to ensure the high-quality of our benchmark and avoided the use of any automated filtering.
>
> **(Q6) Contexts Length:** We have added Table 5 to Appendix A that shows the length distribution of each masked context type for all languages.
>
> **(Ethics) Collecting data from X**: We clarify that we did not perform automated crawling of tweets from X. Rather, we manually searched X using the search bar tool on the website to identify suitable tweets to create our masked contexts for evaluation. We do not release any raw tweets and ensure the contexts we created do not reflect any toxic or harmful content. We have mentioned this more clearly in our ethics statement.
>
> The mentioned changes can be seen in the updated PDF of the submission. We hope that our response and modifications addressed your concerns and would really appreciate it if you could increase your score!

---

> > ### Comment · Reviewer_v7VU · 2025-11-24
> >
> > Thank you for your response, clarifications and revisions.
> >
> > W1 Methodology: I still don't sufficiently understand where the work innovates in terms of methodology. It would be helpful to highlight exactly where innovation was needed to expand Camel to Asian languages, so that others who might want to replicate it for other languages have better guidance.
> >
> > Q1 Data: Thanks for the clarifications of where the data was sourced from. In the revised version it sounds like it was one author who speaks all languages, "For each of the 9 Asian languages, one of our authors who is a native speaker manually filtered".
> >
> > Q3 Llama 3 and Aya Expanse both report the list of supported languages, which exclude some of which you evaluate, e.g. Urdu. Please take these unsupported languages out of the averages, and/or mark them appropriately. If a LLM does officially not support a language, it should not be benchmarked against other models who support it, it is not fair. It is also interesting to discuss what kind of influence it would have on your scores. If the model doesn't support the language, i.e. quality is terrible, would that "debias" it because we can't measure the bias properly?

---

> > > ### Author Response · Authors · 2025-11-25
> > > **Response to Reviewer v7VU**
> > >
> > > Thank you so much for following up on our response!
> > >
> > >
> > > **W1 Methodology:** Thank you for clarifying your concern! There were indeed additional challenges faced when expanding the Arabic Camel benchmark to different Asian languages. While we followed the same general methodology for benchmark design and data collection as Camel, some language-specific challenges required alternative ways for collecting data. We have discussed these specific details in Section 3.2 (language-specific challenges), which we now point to at the beginning of our Section 3. For example, in some cultures such as Chinese, Korean, and Japanese, named entities can be subject to temporal changes and existing knowledge bases like Wikidata do not reflect up-to-date naming conventions, which require us to collect these entities from different sources (i.e., recent governmental reports). We also discussed how certain entity types are not as relevant to all cultures and how we performed tailored adaptations for some entity types. Additionally, since we cover a variety of languages, we performed additional steps of parallelizing Western entities across all languages to enable better comparisons. We hope this discussion provides more clarity on the innovations that were needed when expanding to more languages and serves as additional guidance for future work!
> > >
> > > **Q1 Data:** We clarify that we had 9 total native speakers involved in data collection and annotation (1 person for each language). We have rewritten this sentence to be “The annotation was conducted by nine different authors in total, each a native speaker of one of the 9 Asian languages in Camellia”. Please let us know if this is still confusing!
> > >
> > > **Q3 Language Support:** We clarify that the list of 8 supported languages mentioned by Llama reflects the languages in which instruction-tuning and preference-tuning for helpfulness and safety was done during the post-training phase. However, the model is trained on a much broader range of  176 languages during the pre-training phase, which is mentioned in their technical report (Grattafiori et al. 2024). The huggingface model card also mentions “Llama 3.3 has been trained on a broader collection of languages than the 8 supported languages. Llama may be able to output text in other languages than those that meet performance thresholds for safety and helpfulness.” We found the model to perform reasonably well for the languages we consider on our tasks with the quality not being terrible, even if it hasn’t undergone additional post-training in all of those languages. For example, it can be seen in our results of Figure 6 which shows how Llama achieves accuracies in the 75-90% range for Chinese, Korean, and Japanese, despite them not being in the list of supported languages. This is also the case with the other models we experiment with such as Aya, Gemma, and Qwen. There are other models which were explicitly not trained on other languages besides English (such as Olmo, Phi, and GPT-oss), and for which the quality was indeed terrible, thus we did not include such models in our tasks. Given the lack of transparency by model providers on their pre-training data, it would be unfortunately difficult to know exactly how much coverage of each language a model was trained on. We hope this provides clarification regarding language support, we would be happy to discuss such details in more depth in the appendix. Please let us know if you still have a concern!
> > >
> > > Grattafiori, Aaron, et al. "The llama 3 herd of models." arXiv preprint arXiv:2407.21783 (2024).
> > > https://huggingface.co/meta-llama/Llama-3.3-70B-Instruct

---

### Official Review · Reviewer_JUk5 · 2025-10-31

**Soundness:** 2
**Presentation:** 3
**Contribution:** 3
**Rating:** 4
**Confidence:** 4

**Summary:**

This paper introduces Camellia, a benchmark for measuring entity-centric cultural biases in large language models (LLMs) across six Asian cultures and nine Asian languages, including Chinese, Japanese, Korean, Vietnamese, Urdu, Hindi, Malayalam, Marathi, and Gujarati. The benchmark consists of manually annotated cultural entities and naturally occurring masked contexts from social media. Using Camellia, the authors evaluate four multilingual LLM families (Llama, Qwen, Aya, Gemma) across three tasks: cultural context adaptation, sentiment association, and extractive QA. Results highlight cultural biases.

**Strengths:**

The paper addresses a timely problem of how LLMs treat culturally grounded entities. Camellia is large-scale, contains manually curated annotations, and covers diverse Asian cultures, filling a gap in multilingual fairness resources. The benchmark design is thoughtful and includes English translations to enable cross-lingual comparisons. Experiments are systematic and reveal insights:
(1) models often fail at cultural adaptation,

(2) different model families show opposite sentiment biases, and

(3) QA performance gaps shrink when switching to English.

The discussion of data-collection challenges is detailed and transparent. Overall, the benchmark has strong potential for community impact.

**Weaknesses:**

Some methodological details could be more deeply analyzed. While cultural bias is measured by comparing Asian vs Western entities, Western entities are parallel across languages, potentially oversimplifying cultural grounding. Benchmark construction depends heavily on social-media contexts which raises domain-coverage concerns and may not represent broader text ecosystems. Several analyses rely on probability-based metrics (e.g., CBS) without exploring sensitivity to tokenization granularity. The authors speculate that training-data provenance explains results; however, little direct evidence supports this. The work would benefit from stronger causal attribution and more diagnostic error analysis.

The paper covers a few "asian" languages, but could do a better job of exploring how language relatedness and differences affect results. E.g., is this a typologically diverse language sample, and how do typological differences affect the results.

**Questions:**

Data Representativeness: How representative is your data in a broader cultural discourse? Did you test whether cultural-adaptation findings transfer to news or conversational corpora?

Tokenization Effects: Since CBS depends on token probabilities, how sensitive are results to tokenization differences across languages?

Typological Diversity:  What is the typological diversity of your language sample, and how do language differences affect your results?

---

> ### Author Response · Authors · 2025-11-22
> **Response to Reviewer JUk5**
>
> Thank you so much for your efforts reviewing our paper, we really appreciate it! We respond to your concerns and questions below.
>
> **(W1) Parallelization of Western entities:** We clarify that the reason behind parallelizing the Western entities across all languages is to enable consistent comparisons in our experiments. The Western entities that can be collected in each individual language can vary, with lower-resource ones (Malayalam, Marathi, Urdu, Gujarati) having less mentions of Western entities. To have the same representation of Western entities in all languages, we performed the parallelization procedure which required a lot of effort for manual translation.
>
> **(W2) Use of social media contexts:** We believe that social media contexts are suitable for our evaluation setup, since our objective is to construct simple but natural masked contexts to evaluate LLMs (such as the examples in Figure 1). Social media posts often reflect linguistic patterns of how native people talk and contain discussions mentioning cultural entities, which are important to construct the culturally-grounded contexts in Camellia. Additionally, compared to other text domains such as news, legal, or medical, social media posts are often short and have a clear sentiment which are important properties for our experiments in sentiment association where the simple swap of entities in the [MASK] impacts model behavior. We have included a list of examples of the contexts from the dataset in Appendix A (Figures 9 and 10). We hope this clarifies the reasoning behind our decision to construct masked contexts from tweets!
>
> **(W3) Tokenization impact on CBS:** This is a really interesting point! We have run an additional analysis that we include in Appendix C.1 (Figure 13) that compares the CBS results between Llama3.3-70b and Qwen2.5-72b vs how much the tokenizer vocabulary of each model covers the writing script of each language. The results show that tokenizers with better script coverage for a language (more vocabulary tokens in that script) tend to yield a better CBS. For example, Qwen has better coverage for Chinese, Korean, and Japanese, leading to better performance. On the other hand, Llama has better coverage for Urdu and most Indian languages than Qwen, which leads to better performance on those languages.
>
> **(W4)  Impact of Training Data:** We agree that training data is not the only factor that impacts the results in our experiments. However, we believe it is an important part of many confounding factors such as tokenization differences. We hope that our above analysis on tokenization differences further strengthens our hypothesis. Specifically, better script coverage in a tokenizer (e.g., more coverage of Chinese script in Qwen) naturally reflects a higher proportion of that language in the training data, since these vocabulary tokens are added during tokenizer training based on frequency. Unfortunately, since the pre-training data of these multilingual models are not open-sourced, it would be difficult to perform further pretraining data analysis. We have clarified this in our writing and we hope that with more open-source releases in the future, our benchmark will enable these types of analyses!
>
> **(W5) Typological Diversity:** We have included a description of the typological diversity of the set of languages in Camellia in Appendix A (Table 6). In short, we cover a rich diversity of languages that belong to 6 language families, 4 types of morphologies, and 8 writing scripts. Studying how typological relatedness affects performance of different languages is indeed an interesting direction! However, this type of analysis would unfortunately require a much larger coverage of languages than the set of 9 languages we study in Camellia. For example, the recent work of Ploeger et al. (2024) on typological diversity in NLP uses around 50 languages to conduct their analyses.
>
> Ploeger, Esther, et al. "What is” Typological Diversity” in NLP?." EMNLP 2024.
>
> The mentioned changes can be seen in the updated PDF of the submission.  We hope that our response and modifications addressed your concerns and would really appreciate it if you could increase your score!

---

### Official Review · Reviewer_EX4b · 2025-11-01

**Soundness:** 3
**Presentation:** 3
**Contribution:** 2
**Rating:** 6
**Confidence:** 2

**Summary:**

Camellia is a multilingual benchmark to measure entity-centric cultural biases in LLMs across 9 Asian languages covering 6 Asian cultures. It contains 19,530 manually annotated cultural entities (Asian- vs Western-associated) and 2,173 naturally occurring masked contexts sourced from native-speaker posts on X/Twitter, organized for three evaluations: cultural context adaptation, sentiment association, and extractive QA. The benchmark also provides English translations of entities/contexts.

Evaluation setup: The authors test four recent multilingual LLM families (Llama, Qwen, Aya, Gemma) using prompting (no task-specific fine-tuning) and log-prob–based scoring for masked fills. A central metric, the Cultural Bias Score (CBS), counts the fraction of pairwise cases where a Western entity receives higher likelihood than the culturally appropriate Asian entity in a given context (lower is better)

**Strengths:**

**s1: Originality**
Builds on Arabic-only prior work by scaling to 6 Asian cultures / 9 languages with English parallels; introduces an entity-centric design plus a thoughtful task triad (context adaptation, sentiment, extractive QA) and a clear, reproducible CBS metric.

**s2: Quality**
Uses native, naturally sourced contexts; good scale (~19.5k entities, ~2.2k contexts); high inter-annotator agreement; and unified prompting/decoding across strong LLM families for fair, controlled comparisons.

**Weaknesses:**

**w1: Flow/structure**
The write-up feels a bit scattered; a short “where we fit vs prior work” in the intro and a 3–4 bullet “Key Contributions” box would make the novelty and takeaways pop. (Style preference, but it really helps readability.)

**w2: One scoreboard (nice to have, not required)**
A single at-a-glance table ranking models across tasks/languages would make comparisons easier, but it’s optional.

**w3: Simple control with English-only models**
Since English parallels exist, running strong monolingual English models would help separate language issues from cultural knowledge.

**Questions:**

**Release package**
Will you release the prompts, and scripts necessary to reproduce the main tables?

**Overall**
well-constructed dataset and a useful benchmark. Good coverage (6 cultures/9 languages), and a clear evaluation protocol make this a valuable resource. The paper is generally clear, and the analyses are informative. My main suggestion to strengthen the work is to test a broader set of models (including strong monolingual English models on the English parallels) to sharpen conclusions and improve comparability.

---

> ### Author Response · Authors · 2025-11-22
> **Response to Reviewer EX4b**
>
> Thank you so much for your efforts reviewing our paper, we really appreciate it! We respond to your concerns and questions below.
>
> **(W1)  Flow/Structure:** We have incorporated your suggestion and rewrote our introduction to include 3-4 bullet points that summarize our key contributions and improve clarity. Thank you for this point!
>
> **(W2) Leaderboard:** We have added leaderboards for each of our tasks in Appendix C.4 which show the best performing model when testing in all languages and in English. Thanks for this great suggestion to summarize our results!
>
> **(W3) Comparisons with English monolingual models:** We have run additional experiments with the Olmo2-32b and Phi-4 models which are trained only on English data and included their results in our additional experiments in English in Appendix C. The results are also added to our new leaderboards. We find that in some cases models trained on English only data perform better than ones trained with additional languages.
>
>
> **(Q1) Data & Code Release:** Yes, we will publicly release our data and scripts to run tasks with the community.
>
>
> The mentioned changes can be seen in the updated PDF of the submission. We hope that our response and modifications addressed your concerns and would really appreciate it if you could increase your score!

---

### Author Response · Authors · 2025-12-03
**Summary for AC**

Dear AC, thank you for your effort in reviewing our submission! We provide a summary of our discussion with the reviewers.

The paper received many valuable suggestions from all 4 reviewers. Our scores were increased from (4, 4, 6, 6) to (4, 6, 6, 6) after an engaging discussion with reviewers who read our responses and revisions to the paper, before the rebuttal ended due to the leak.

We summarize the main updates we made to the paper below which address the reviewers’s concerns:

- *Structure and Writing Clarifications:* We made several improvements to the flow of the paper by re-organizing the placement of our discussion of language-specific details and our related work discussion. We also made several writing modifications that clarify the definition of Asian and Western cultures in our paper and some data collection details. We also rephrased some sentences according to the suggestion of the reviewers.

- *Additional Dataset Details:*  We included an additional section that provides a detailed overview of the typological diversity of our dataset which covers 9 different Asian languages. We also reported additional details such as the length distribution of all the contexts in our dataset, and the distribution of Western entities across Western countries. Further, we performed additional post-annotation quality checks with new external annotators for all languages that further confirmed the quality of our dataset.

- *New Tokenization Analysis:* We included an additional analysis on the impact of tokenization differences between LLMs for cultural adaptation. This new analysis further strengthened our results when comparing models developed in various regions.

- *Additional Model Comparisons and Scoreboards*: We included additional results of monolingual English LLMs and reported multiple scoreboards that compare all models on all languages and tasks in our benchmark.

We hope that the score increase we had and the mentioned satisfaction of the reviewers with the changes we made would be taken into account. Thank you again for your time and service!

---

### Meta-Review · Area_Chair_qFBg · 2025-12-22

**Summary:**

This resource paper falls in a borderline area, originally, there were multiple issues with the writing, some of which might have been addressed during the rebuttal period. Moreover, the paper deals with a topic that raises excitement and the value of the resouce was acceptted by all reviewers. Still, both reviewers that leaned positive and negative had issues with the thought and explanation concerning the methodology.

**Reviewer Concerns:**

Writing, related work and such efforts are never a box to check. As there were many suggestions for improvement a few iterations on that are advised. Moreover, the methodological concerns, the replicability details etc. should be considered thorowly and discussed or changed as needed.
Note that you have a type in the tl;dr (nice).

**Reviewer Scores:**

That is not a fair, relevant or meaningful question. I protest the way this was all handled.
A Reviewers are not here, and ToM is weak, at least mine and the one literature study. I will not try to predict people.
B Scores are, anyway, a weak signal of interest; a paper should not be accepted or rejected just based on it. An AC's job is to look at the specific weaknesses and translate them into a recommendation. (even here there are two reviews with similar issues, one leaning accept one reject, but both weakly so)
C There are about 100 pages of discussions for me to read overall, in addition to the discussions I monitored and were just replaced, this is beyond my personal ability to do fairly. I did my best effort.

---

### Decision · Program_Chairs · 2026-01-26

Reject